**There are amendments to this paper**

# MiR-31 promotes mammary stem cell expansion and breast tumorigenesis by suppressing Wnt signaling antagonists

Cong Lv[1], Fengyin Li[1], Xiang Li[1], Yuhua Tian[1], Yue Zhang[2], Xiaole Sheng[1], Yongli Song[1], Qingyong Meng[1], Shukai Yuan[3], Liming Luan[4], Thomas Andl[4], Xu Feng[5], Baowei Jiao [5], Mingang Xu[6], Maksim V. Plikus[7], Xing Dai[8], Christopher Lengner[9,10,11], Wei Cui [1,12], Fazheng Ren[1], Jianwei Shuai[13], Sarah E. Millar[6,11] & Zhengquan Yu [1]

MicroRNA-mediated post-transcriptional regulation plays key roles in stem cell self-renewal and tumorigenesis. However, the in vivo functions of specific microRNAs in controlling mammary stem cell (MaSC) activity and breast cancer formation remain poorly understood. Here we show that miR-31 is highly expressed in MaSC-enriched mammary basal cell population and in mammary tumors, and is regulated by NF-κB signaling. We demonstrate that miR-31 promotes mammary epithelial proliferation and MaSC expansion at the expense of differentiation in vivo. Loss of miR-31 compromises mammary tumor growth, reduces the number of cancer stem cells, as well as decreases tumor-initiating ability and metastasis to the lung, supporting its pro-oncogenic function. MiR-31 modulates multiple signaling pathways, including Prlr/Stat5, TGFβ and Wnt/β-catenin. Particularly, it activates Wnt/β-catenin signaling by directly targeting Wnt antagonists, including Dkk1. Importantly, Dkk1 overexpression partially rescues miR31-induced mammary defects. Together, these findings identify miR-31 as the key regulator of MaSC activity and breast tumorigenesis.

[1] State Key Laboratories for Agrobiotechnology and Beijing Advanced Innovation Center for Food Nutrition and Human Health, College of Biological Sciences, China Agricultural University, Beijing 100193, China. [2] Department of Biochemistry and Molecular Biology, Hebei Key Laboratory of Chinese Medicine Research on Cardio-Cerebrovascular Disease, Hebei University of Chinese Medicine, Shijiazhuang, Hebei 050200, China. [3] Department of Biochemistry and Molecular Biology, Basic Medical College, Tianjin Medical University, Tianjin 300070, China. [4] Vanderbilt University Medical Center, Nashville, TN 37232, USA. [5] State Key Laboratory of Genetic Resources and Evolution of Kunming Institute of Zoology, Chinese Academy of Sciences, Kunming, Yunnan 650223, China. [6] Department of Dermatology, Perelman School of Medicine, University of Pennsylvania, Philadelphia, PA 19104, USA. [7] Department of Developmental and Cell Biology, Sue and Bill Gross Stem Cell Research, Center for Complex Biological Systems, University of California, Irvine, CA 92697, USA. [8] Departments of Biological Chemistry and Dermatology, School of Medicine, University of California, Irvine, CA 92697, USA. [9] Department of Biomedical Sciences, School of Veterinary Medicine, University of Pennsylvania, Philadelphia, PA 19104, USA. [10] Institute for Regenerative Medicine, University of Pennsylvania, Philadelphia, PA 19104, USA. [11] Department of Cell and Developmental Biology, Perelman School of Medicine, University of Pennsylvania, Philadelphia, PA 19104, USA. [12] Institute of Reproductive and Developmental Biology, Department of Surgery and Cancer, Imperial College London, London W12 0NN, UK. [13] Department of Physics and State Key Laboratory of Cellular Stress Biology, Innovation Center for Cell Signaling Network, Xiamen University, Xiamen 361005, China. Cong Lv, Fengyin Li, Xiang Li and Yuhua Tian contributed equally to this work. Correspondence and requests for materials should be addressed to Z.Y. (email: zyu@cau.edu.cn)

Mammary gland is a unique organ in that major developmental changes, including ductal morphogenesis, side tertiary branching and alveogenesis, occur postnatally[1]. The mammary epithelia exhibit a relatively simple lineage composition with luminal cells capable of terminally differentiating into milk-producing cells, and basal/myoepithelial cells that possess some mesenchymal-like features[2]. Mammary gland development and homeostasis are fueled by multipotent mammary stem cells (MaSCs), as well as unipotent stem/progenitor cells[3, 4]. A mammary epithelial cell population enriched for MaSCs has been isolated from the basal compartment based on their expression of CD24 and CD29 or CD49f antigens[5, 6]. Wnt targets such as Axin2, Procr and Lgr5, which are specifically expressed in basal MaSCs, have been used to identify distinct Wnt-responsive MaSC subsets[7–9]. Striking a balance between MaSC self-renewal and differentiation is essential to maintain mammary tissue homeostasis. Elucidating the molecular mechanisms that govern this balance is critical for understanding the basic principles of mammary development and the ontogeny of breast cancer.

MaSCs are controlled by the dynamic interplay of multiple molecular pathways such as hormone, Notch and Wnt signaling[7, 10, 11]. Progesterone/PR(progesterone receptor) plays a prominent role in promoting the proliferative capacity of MaSCs and coordinating alveogenesis during early pregnancy via secreted RANKL[11, 12]. RANKL binds to its receptor RANK and activates NF-κB signaling in myoepithelial cells[13, 14]. In addition to regulating MaSC activity and alveologenesis in normal mammary epithelia, RANKL and RANK are critical for the maintenance of cancer stem cells and for breast cancer metastasis[15]. Interestingly, RANKL and RANK are predominantly expressed in hormone receptor-negative, but not receptor-positive, human breast tumors[15–17], raising the possibility of their activation by hormone-independent mechanisms in malignant mammary epithelia. Wnt/β-Catenin signaling is important for promoting MaSC activity and determining a basal cell fate. Wnt ligands such Wnt4 and Rspo1 have been identified as the niche factors for MaSCs, functioning to promote MaSC self-renewal[18, 19]. Forced activation of Wnt signaling in *MMTV-Wnt1* and *MMTV-stabilized-β-Catenin* transgenic mice expands mammary stem/progenitor cell populations[5, 20–22]. Moreover, hyperactive Wnt signaling is extensively presented in breast cancer, particularly in basal-like type with higher grade, stem cell-like characteristics and aggressive behavior[23]. Although the involvement of Wnt/β-Catenin signaling in MaSC biology and breast cancer has been extensively studied, how it is precisely controlled in mammary gland to balance stem cell self-renewal and differentiation remains to be fully understood.

MicroRNAs have been shown to play important roles in controlling adult stem cell fate and tumorigenesis[24]. Specifically, *miR-31* has been identified as an important regulator of adult muscle and mesenchymal stem cells[25–27]. Several reports showed that *miR-31* is enriched in putative mammary progenitor cells[28–30]. *MiR-31*'s importance has also been implicated in a variety of cancers including breast cancer[31]. However, the in vivo function of *miR-31* in mammary gland development, MaSC activity and breast tumorigenesis remain unknown. By utilizing *miR-31* gain- and loss-of-function mouse models, coupled with the *MMTV-PyVT* mammary tumor model, here we demonstrate that *miR-31* promotes MaSC activity and breast tumorigenesis by regulating multiple signaling pathways.

## Results

### *MiR-31 is enriched in MaSC population and breast tumors*. To identify the mammary epithelial cell populations that express *miR-31* in vivo, we purified Lin⁻CD24⁻CD29⁻, Lin⁻CD24⁻CD29⁺, Lin⁻CD24⁺CD29low and Lin⁻CD24⁺CD29high subpopulations, confirming their purity by the expression of basal marker K14 and luminal marker K18 (Supplementary Fig. 1a). Mature *miR-31* was highly enriched in the CD24⁺CD29high cell population, with lower level of expression in the other populations (Fig. 1a). This pattern parallels that of other MaSC-enriched microRNAs, *miR-205* and *miR-22*[28, 32, 33]. In situ hybridization revealed the presence of *miR-31*-expressing cells in both basal and luminal layers, with a particular enrichment in the tertiary branches (Fig. 1b).

During mammary development, *miR-31* expression gradually increased from puberty to adult stages, peaking around post-pregnancy day 14.5 (14.5 d.p.c.), and then returning to pre-pregnancy levels upon involution (Fig. 1c). This dynamic expression pattern is similar to that of NF-κB[34], suggesting a potential correlation between *miR-31* and NF-κB. To probe this, we examined the *miR-31* promoter using the JASPAR database, and identified two potential NF-κB binding sites at positions −1,746 and −1,375 (Fig. 1d). RANKL is an activator of the NF-κB pathway[14]. Knockdown of RANKL with siRNA repressed *miR-31* expression (Fig. 1e), concomitant with repression of the NF-κB pathway (Fig. 1f). In *miR-31* promoter driven-luciferase reporter assays, both RANKL siRNA and mutation of p65-binding sites in *miR-31* promoter suppressed luciferase activity (Fig. 1g). Furthermore, chromatin immunoprecipitation (ChIP) assay revealed that p65 binds to its predicted cognate sites in *miR-31* promoter, and that RANKL siRNA reduced this binding (Fig. 1h). Together, these data suggest that *miR-31* expression is directly activated by NF-κB pathway in the mammary gland.

Given that the NF-κB pathway is activated in mammary myoepithelium by RANKL secreted from the luminal epithelium upon progesterone signaling[35], we asked if *miR-31* expression is also regulated by progesterone. Treatment with estradiol and progesterone significantly, albeit moderately, increased *miR-31* expression (Supplementary Fig. 1b), whereas PR inhibitor Mifeprinstone decreased *miR-31* expression (Supplementary Fig. 1c-e). In *miR-31* promoter driven-luciferase reporter assays, treatment of estradiol and progesterone enhanced luciferase reporter activity, while mutation of p65-binding sites blocked it (Supplementary Fig. 1f). Moreover, treatment of estradiol and progesterone also enhanced p65 recruitment to its binding site while Mifeprinstone repressed it (Supplementary Fig. 1g). Thus, *miR-31* expression is also hormone-responsive, likely through an indirect mechanism that involves the RNAKL/RANK/NF-κB pathway.

Next we examined *miR-31* expression in mammary tumors. *MiR-31* was markedly elevated in *PyVT*-induced tumors (Fig. 1i), which at late stage expressed high level of RANKL despite an ER- and PR-negative appearance (Fig. 1k and Supplementary Fig. 1h)[15], as well as high levels of p-p65 and p-IKKα (Fig. 1j). Akt is activated in *PyVT* tumors[36] and NF-κB is activated through the Akt-IKKα pathway[37, 38], raising the possibility that *miR-31* is induced by Akt pathway activation. Indeed, treatment with Pten inhibitor bpV induced *miR-31* expression in a dose-dependent manner, concomitant with upregulation of p-Akt, p-Ikkα and p-p65 (Fig. 1l, m). Strikingly, we also observed an overexpression of *miR-31* in basal-like human breast cancer (Supplementary Fig. 1i), and its level positively correlated with RANKL and TNFα (Supplementary Fig. 1j), both of which are activators of the NF-κB pathway[39, 40]. Thus, *miR-31* induction in mammary tumors is tightly associated with the NF-κB pathway, which might be activated by progesterone/PR-independent mechanisms such as the RANKL and Akt pathways.

### *MiR-31 induction causes hyperplasia*. To investigate the function of *miR-31* in regulating mammary development, we generated *TRE-miR31* mice and bred with *K5-rtTA* mice to generate

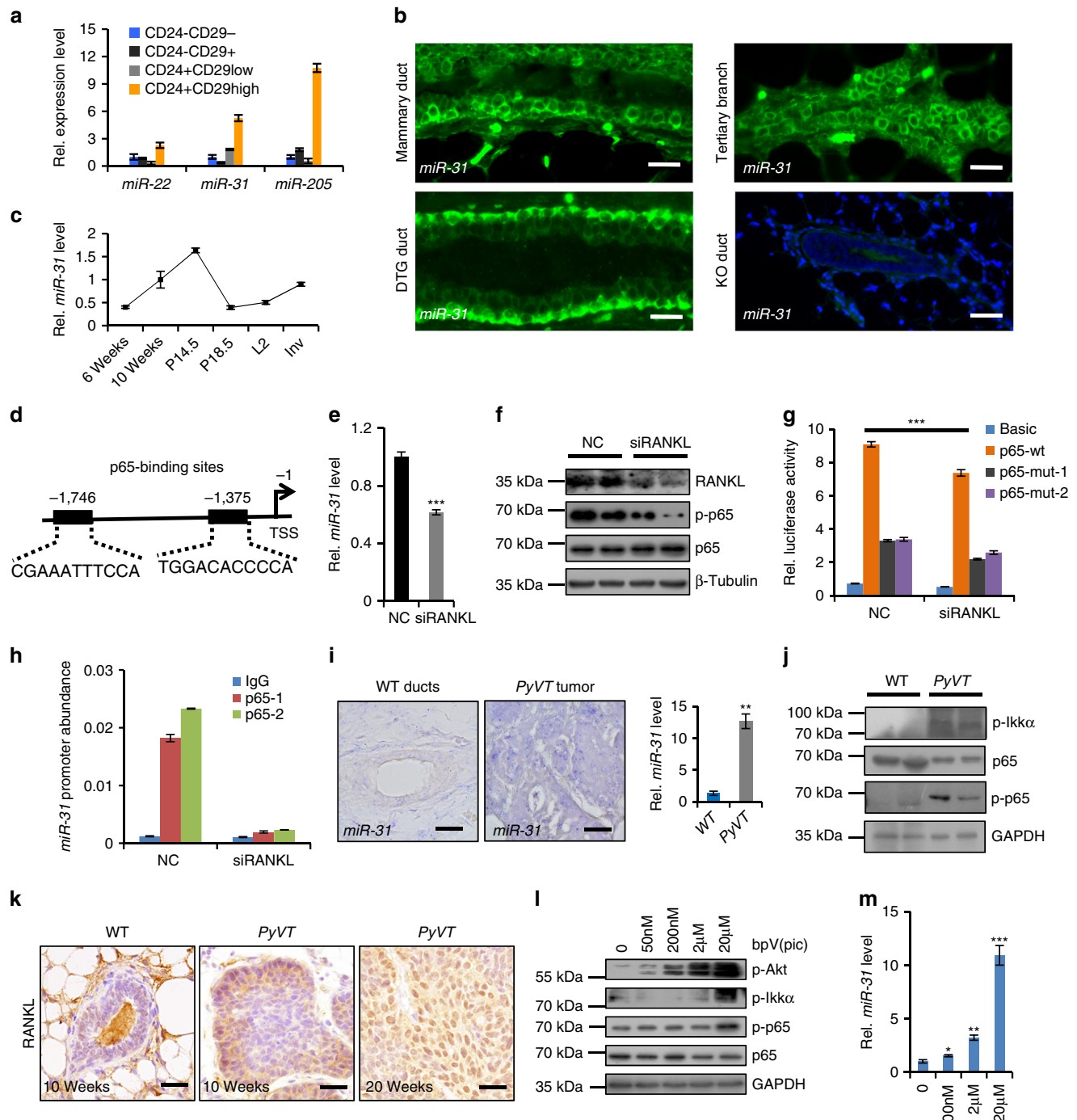

**Fig. 1** Expression pattern of *miR-31* in mammary gland and tumors. **a** qRT-PCR for *miR-31*, *miR-22* and *miR205* in Lin⁻CD24⁺CD29^high, Lin⁻CD24⁺CD29^low, Lin⁻CD24⁻CD29⁺ and Lin⁻CD24⁻CD29⁻ populations at 12 weeks of age. *n* = 3 biological replicates. **b** In situ hybridization for *miR-31* in 12-week-old WT mammary gland ducts and tertiary branches. DTG mammary ducts, a positive control. The DTG mice have been administered with Dox at 1 week of age. *miR-31* KO mammary ducts, a negative control. Scale bar, 25 μm. **c** qRT-PCR for *miR-31* in WT mammary epithelial cells at 6, 10 weeks, P14.5 (14.5 days post pregnancy), P18.5, L1 (1 day post lactation) and Inv (10 days post involution). *n* = 3 at each time points. **d** The schematic diagram showing two potential p65 (NF-κB) binding sites (-1746 bp and -1375 bp) in the *miR-31* promoter. TSS, transcription start site. **e, f** qRT-PCR for *miR-31* **e** and western blotting for RANKL, p-p65, p65 **f** in HC11 mouse mammary epithelial cells treated with RANKL siRNA (SiRANKL), and scramble RNA (NC). **g** Luciferase activity in lysates of HC11 mouse mammary epithelial cells transfected with luciferase reporter plasmids of pGL3-basic plasmid, *miR-31* promoter or *miR-31* mutant promoter with mutation at the -1375 (p65-mut-1) or -1746 (p65-mut-2) binding site, treated with scramble RNA (negative control, NC) and RANKL siRNA. **h** ChIP assays carried out on HC11 mammary epithelial cells using antibodies against p65 under indicated conditions. **i–k** *In situ* hybridization and qRT-PCR analysis for *miR-31* **i** and western blotting for p-Ikkα, p-p65 and p65 **j**, and immunohistochemistry for RANKL **k** in WT mouse mammary gland and *PyVT* tumors. Scale bar, 25 μm. **l** western blotting for p-Akt, p-Ikkα, p-p65 and p65 in MCF7 breast cancer cells treated with Pten inhibitor bpV(pic) at indicated concentrations for 12 h. **m** qRT-PCR for *miR-31* in MCF7 breast cancer cells treated with PTEN inhibitor bpV(pic) at indicated concentrations for 12 h. Data represented as mean ± S.D. Sample size: WT (*n* = 3) and *PyVT* (*n* = 3) for **i–k**. Two tailed unpaired *t*-test for **e, g, i, m** (*P < 0.05; **P < 0.01; ***P < 0.001)

double transgenics (DTG) that inducibly overexpress *miR-31* specifically in mammary basal epithelial cells (Supplementary Fig. 2a). Doxycycline (Dox)-mediated induction of *miR-31* was confirmed (Supplementary Fig. 2b, c), while no significant difference was found for *miR-205*, an unrelated microRNA (Supplementary Fig. 2b). To determine the effects of *miR-31* overexpression on mammary morphology, we induced its

expression starting at 1 week of age by oral administration of Dox-containing water in the lactating females. Following weaning, pups were maintained on Dox-containing water until the end time point. Since Dox itself may affect mammary development, we compared mammary gland morphology from mice fed with either Dox or control water. We found that Dox administration mildly inhibited mammary branching and ductal elongation at

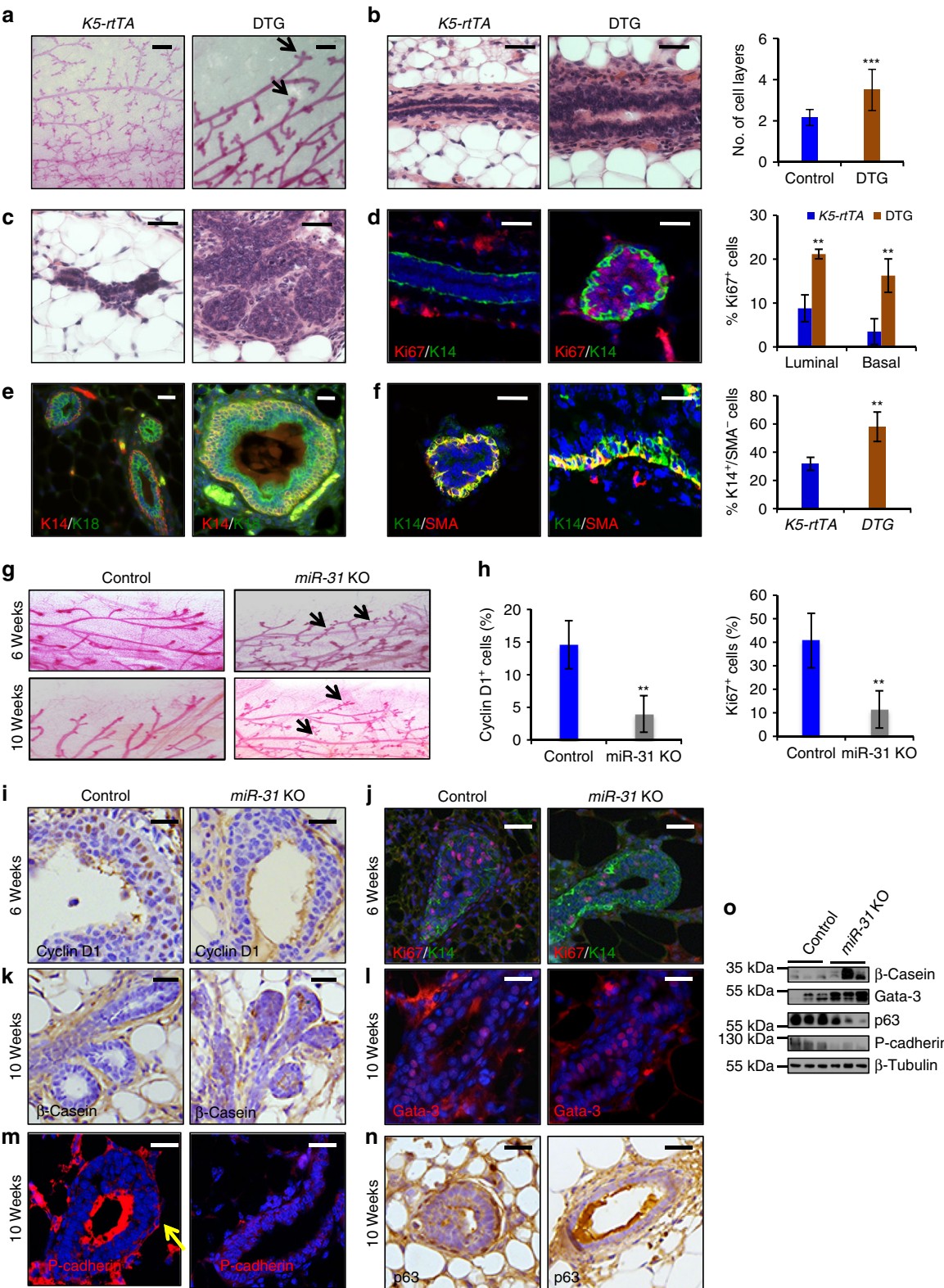

6 weeks of age, but by 10 weeks of age such effects were no longer observed. Therefore, all of the following studies were carried out on Dox-treated 12-week-old female mice.

By 12 weeks of age, mammary glands of all control mice, including WT, *K5-rtTA* and *TRE-miR31*, were fully developed with tertiary branching (Fig. 2a and Supplementary Fig. 2d–f). In contrast, mammary glands of all DTG mice showed no tertiary branch development (Fig. 2a and Supplementary Fig. 2g–j) and varying defects in ductal elongation (Supplementary Fig. 2g–i), branching (Supplementary Fig. 2h–l), and the appearance of multiple terminal end bud (TEB)-like structures (Fig. 2a and Supplementary Fig. 2k). On histology, DTG mice had thicker mammary ducts than controls, characterized by an increase in cell layers (Fig. 2b and Supplementary Fig. 2m), particularly pronounced at the end of the branches (Fig. 2c). Consistent with epithelial hyperplasia, DTG mammary glands exhibited a robust increase in cell proliferation (Fig. 2d). The effect of *miR-31* on proliferation is likely epithelial cell-autonomous, as Dox-induced *miR-31* overexpression in HC11 cells, a mouse mammary epithelial cell line, dramatically promoted proliferation (Supplementary Fig. 2n,o). Conversely, HC11 cell proliferation was significantly repressed in response to *miR-31* inhibitor, *anti-miR-31* (Supplementary Fig. 2o).

The preferential expression of *miR-31* in mammary basal cells led us to examine its influence on basal and luminal lineages. In normal adult mammary glands, ducts are comprised of a single layer of flattened K14+ myoepithelial cells and a single layer of K18+ luminal cells (Fig. 2e). By contrast, DTG mammary ducts contained multiple layers of rounded K14+ cells and displayed an expansion of K18 expression into the basal cell layers, resulting in the appearance of K14/K18-double positive cells (Fig. 2e). Furthermore, in control mice, mammary basal cells were mostly K14+SMA+ (SMA, smooth muscle actin, is a marker of differentiated basal cells) (Fig. 2f), whereas the basal layers of DTG mice contained many K14+SMA− cells, suggesting a possible arrest in myoepithelial cell maturation (Fig. 2f). Taken together, these data demonstrate that *miR-31* overexpression causes mammary hyperplasia likely via promoting the proliferation and inhibiting the differentiation of mammary epithelial cells.

***MiR-31* prevents precocious alveolar differentiation.** To probe the physiological function of *miR-31*, we generated a constitutive *miR-31* KO mouse model (Supplementary Fig. 3a–c). The *miR-31* KO mice exhibited normal overall development and were fertile. However, analysis of their mammary glands at 6 weeks of age (puberty) and 10 weeks of age (post-puberty) revealed precocious alveolar differentiation, resembling glands in early pregnancy (Fig. 2g and Supplementary Fig. 3d). A remarkable reduction in the number of proliferating cells was observed in TEBs of *miR-31*

KO mice at 6 weeks of age (Fig. 2h–j). By 10 weeks of age, *miR-31* KO mammary glands expressed milk protein β-Casein, while no appreciable expression was seen in WT mammary glands at this stage (Fig. 2k). Moreover, the number of cells positive for Gata-3, a transcription factor required for differentiation of the luminal epithelium[41], was markedly increased in *miR-31* KO mammary glands (Fig. 2l, o). To determine whether the mammary phenotypes are due to loss of *miR-31* in epithelial cells, we generated *K14-Cre;miR-31^fl/fl* mice in which *miR-31* is specifically deleted in the mammary epithelium (Supplementary Fig. 3e, f). A similar, although slightly milder, phenotype of precocious alveolar differentiation was observed in *K14-Cre;miR-31^fl/fl* mice at 10 weeks of age (Supplementary Fig. 3g), suggesting that *miR-31* functions within the virgin mammary epithelium to keep alveolar differentiation in check.

The precocious alveolar differentiation phenotype in *miR-31* KO virgin mice closely mimics that observed in *P-cadherin* null mice[42]. Consistently, P-cadherin, which is normally expressed in mammary myoepithelial cells, was absent in *miR-31* KO mammary epithelium (Fig. 2m, o). Further, the number of cells positive for p63, which plays a key role in myoepithelial development and in activating P-cadherin expression[43], was markedly reduced in *miR-31* KO mammary epithelium (Fig. 2n, o). These results indicate that the precocious differentiation phenotype in *miR-31* KO mice is likely associated with reduction of p63- and P-cadherin-positive cells. Overall, the *miR-31* KO phenotypes of precocious alveolar differentiation and reduced cell proliferation are directly opposite to these seen in the *miR-31*-overexpressing mice, indicating an important physiological role for *miR-31* in promoting cell proliferation and repressing differentiation.

***MiR-31* regulates mammary alveogenesis.** Next we examined the effect of *miR-31* overexpression on alveologenesis during pregnancy. DTG mice were able to give birth, similar to controls, but were incapable of nursing their pups, leading to neonatal lethality. Whole-mount and histology analysis revealed that DTG mammary glands failed to form alveoli and retained undifferentiated ductal structures at post-pregnancy 14.5 (14.5 d.p.c.) and lactation day 1 (L1) (Fig. 3a, b). Consistently, β-Casein was completely absent in DTG glands (Fig. 3c). *MiR-31* overexpression in HC11 cells, which can be differentiated in vitro upon the addition of lactogenic hormones, also downregulated β-Casein expression (Supplementary Fig. 4a). Conversely, *miR-31* inhibition led to a modest upregulation of β-Casein expression and resulted in lipid formation in this system (Supplementary Fig. 4a). These findings demonstrate that excessive amounts of *miR-31* can repress alveolar differentiation of the mammary epithelium during pregnancy.

**Fig. 2** *MiR-31* promotes cell proliferation and concomitantly represses differentiation of mammary epithelium. **a** Whole mount staining of *K5-rtTA* and DTG mammary glands at 12 weeks of age. Arrows, DTG mammary ductal ends with three or more TEB-like structures. Scale bar, 0.2 mm. **b** H&E staining of *K5-rtTA* and DTG mammary ducts and quantification of cell layers of mammary ductal epithelium. Scale bar, 25 μm. **c** H&E staining of *K5-rtTA* and DTG mammary TEBs. Scale bar, 25 μm. **d** Immunofluorescence for Ki67 in *K5-rtTA* and DTG mammary glands and quantification of Ki67+ cells in basal and luminal layers. Scale bar, 25 μm. **e** K18 and K14 double immunofluorescence in *K5-rtTA* and DTG mammary glands. Scale bar, 25 μm. **f** SMA (red) and K14 (green) double immunofluorescence in *K5-rtTA* and DTG mammary glands. Statistical analysis of K14+/SMA− cells in *K5-rtTA* and DTG mammary glands. **p < 0.01. Scale bar, 25 μm. **g** Whole mount staining of Control and *miR-31* KO mammary glands at 6 weeks and 10 weeks of age. Arrows point to precocious alveoli. **h** Statistical analysis of Cyclin D1- and Ki67- positive cells in TEBs of 4 Control and 4 *miR-31* KO mammary glands at 6 weeks of age in panels **i** and **j**. **i** Immunohistochemistry for Cyclin D1 in TEBs of Control and *miR-31* KO mammary glands at 6 weeks of age. Scale bar, 25 μm. **j** Ki67 (red) and K14 (green) double immunofluorescence in TEBs of Control and *miR-31* KO mammary glands. Scale bar, 25 μm. **k–n** Immunostaining for β-Casein, Gata-3, P-cadherin and p63 in Control and *miR-31* KO mammary glands at 10 weeks of age. Scale bar, 25 μm. **o** Western blotting for β-Casein, Gata-3, p63 and P-cadherin in Control and *miR-31* KO mammary glands at 10 weeks of age. β-Tubulin was used as a loading control. Data represented as mean ± S.D. Sample size: *K5-rtTA* (n = 3) and DTG (n = 3) for **a–f**; Control (n = 3) and *miR-31* KO (n = 3) for **g–o**. Two tailed unpaired *t*-test for **b, d, f, h** (**P < 0.01; ***P < 0.001)

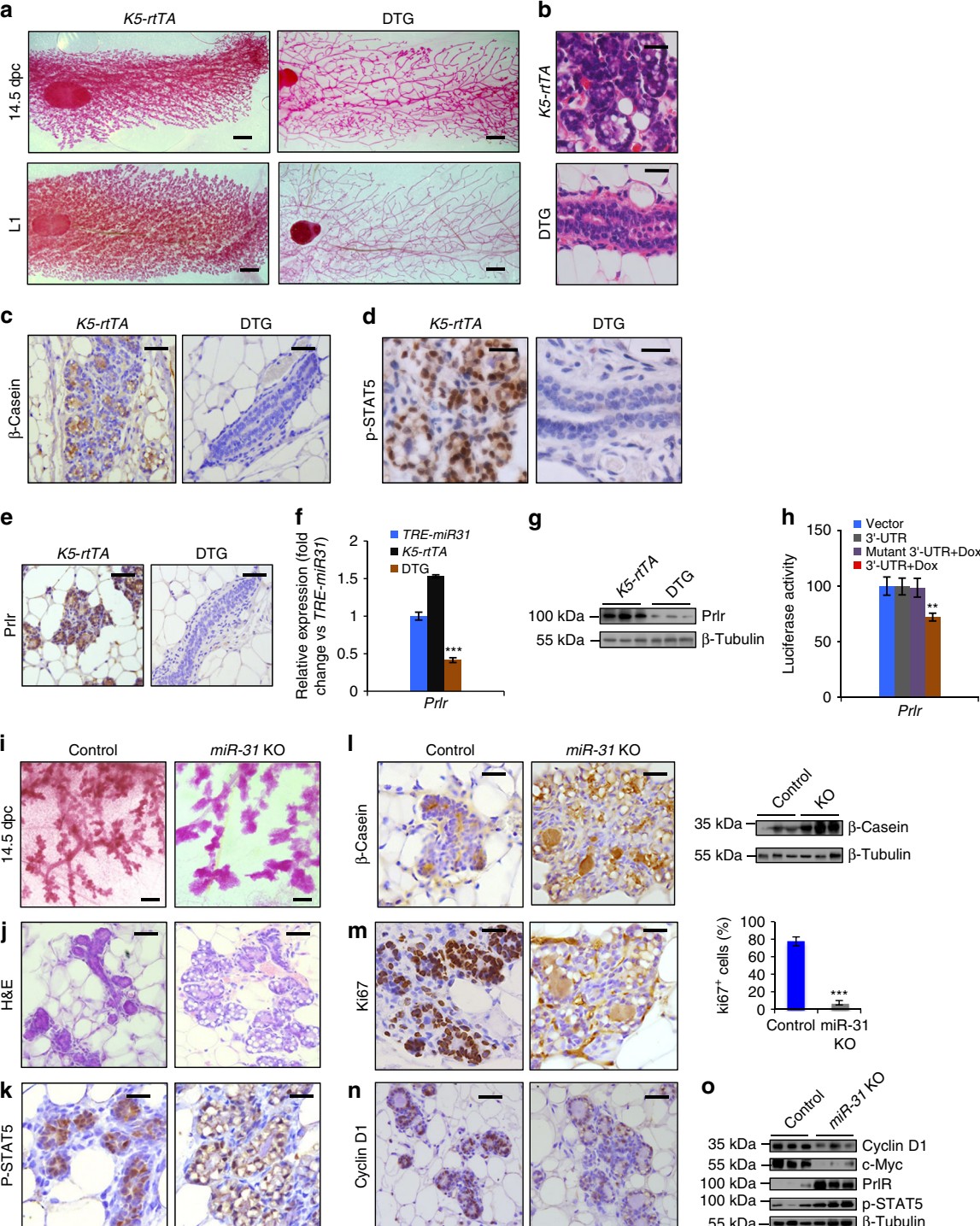

**Fig. 3** *MiR-31* is required for normal alveogenesis during pregnancy. **a** Whole-mount staining of *K5-rtTA* and DTG mammary glands at pregnancy day 14.5 (14.5 d.p.c.) and lactation 1 day (L1) following Dox treatment from 3 weeks of age. $n = 3$ biological replicates at each time points. Scale bar, 1 mm. **b** H&E staining of *K5-rtTA* and DTG mammary glands at 14.5 d.p.c. Scale bar, 25 μm. **c–e** Immunohistochemistry for β-Casein, p-STAT5 and Prlr in *K5-rtTA* and DTG mammary glands at 14.5 d.p.c. Scale bar, 25 μm. **f** qRT-PCR analysis for *Prlr* in DTG mammary epithelium as compared to *K5-rtTA* and *TRE-miR31* mammary epithelium. **g** Western blotting for Prlr in *K5-rtTA* and DTG mammary glands. β-Tubulin was used as a loading control. **h** Luciferase reporter activity of *Prlr* in wild-type and mutant 3′-UTR constructs in *miR-31* inducible HC11 mammary epithelial cells in the presence or absence of Dox (Dox induces *miR-31* overexpression).
**i, j** Whole-mount staining **i** and H&E staining **j** in control and *miR-31* KO mammary glands at 14.5 d.p.c. Scale bar, 0.2 mm **i**; 25 μm **j**. **k** Immunohistochemistry for p-STAT5 in Control and *miR-31* KO mammary glands at 14.5 dpc. Scale bar, 25 μm. **l** Immunohistochemistry and western blotting for β-Casein in Control and *miR-31* KO mammary glands at 14.5 d.p.c. β-Tubulin was used as a loading control. Scale bar, 25 μm. **m** Immunohistochemistry for Ki67 in Control and *miR-31* KO mammary glands at 14.5 d.p.c. and statistical analysis of Ki67+ cells. Scale bar, 25 μm. ***$P < 0.001$. **n** Immunohistochemistry for Cyclin D1 in Control and *miR-31* KO mammary glands at 14.5 d.p.c. Scale bar, 25 μm. **o** Western blotting for Cyclin D1, c-Myc, Prlr and p-STAT5 in Control and *miR-31* KO mammary epithelium. β-Tubulin was used as a loading control. Data represented as mean ± S.D. Sample size: *K5-rtTA* ($n = 3$), *TRE-miR31* ($n = 3$) and DTG ($n = 3$) for **a–g**; Control ($n = 3$) and *miR-31* KO ($n = 3$) for **i–o**. Two tailed unpaired $t$-test for **f**, **h**, **m** (**$P < 0.01$; ***$P < 0.001$)

Prlr/Stat5 signaling is critical for alveolar formation during pregnancy[44]. In contrast to WT alveoli where many p-Stat5-positive cells were detected, such cells were completely absent in DTG mammary glands (Fig. 3d). Prlr-positive cells were also absent in DTG glands (Fig. 3e), and reduction of Prlr was further confirmed at both mRNA and protein levels (Fig. 3f, g). Progesterone signaling is another key regulator of alveolar formation, but the number of PR-positive cells remained unaltered in DTG glands (Supplementary Fig. 4b). Interestingly, the 3′-UTR of Prlr includes a predicted miR-31 binding site (Supplementary Fig. 4c). Luciferase reporter assay showed that elevated miR-31 upon Dox treatment strongly repressed WT Prlr 3′-UTR reporter activity, but had no influence on mutant reporters where the binding site was disrupted (Fig. 3h). Together, these data suggest a selective inhibitory effect of miR-31 on Prlr/Stat5 signaling, and underscore Prlr 3′-UTR as a direct molecular target of miR-31.

We also examined miR-31 KO mice for possible alveologenesis defects. While these mice were fertile, we noticed that by weaning age the survival rates of their pups were lower than those from WT mothers (Supplementary Fig. 4d). The pups nursed by miR-31 KO mothers had less milk in their stomachs at lactation day 2 (Supplementary Fig. 4e). The alveoli failed to fully develop in miR-31 KO mothers at both 18.5 d.p.c. and lactation day 2 (Supplementary Fig. 4f). In line with this observation, the miR-31 KO mammary glands displayed a reduction in lipid droplets and alveolar tissues (Supplementary Fig. 4g). β-Casein immunofluorescence showed that milk content was markedly decreased in miR-31 KO glands (Supplementary Fig. 4h). These findings were somewhat surprising given that we anticipated precocious alveologenesis in pregnant miR-31 KO glands based on data discussed above. To identify the underlying cause of impaired alveogenesis, we examined miR-31 KO mammary glands at mid-pregnancy (14.5 d.p.c.). At this earlier time point, enhanced alveolar content was detected in miR-31 KO glands (Fig. 3i), presenting as overt luminal feature and an increased number of intracellular lipid droplets (Fig. 3j). Expression of β-Casein was also much higher than in control glands (Fig. 3l). Similarly to that observed in the virgin mammary glands, the levels of Prlr and p-Stat5 proteins increased (Fig. 3k, o), and the number of proliferating cells decreased in miR-31 KO alveoli at 14.5 d.p.c. (Fig. 3m–o). Thus, it appears that the primary consequence of miR-31 loss is precocious differentiation of the alveolar epithelium, which is followed by subsequent defect in alveolar development.

***MiR-31 is critical for the self-renewal of MaSCs.*** Next we examine the specific effect of miR-31 on MaSCs. Compared to K5-rtTA and TRE-miR31 control mice, DTG mice at the same estrus stage (Supplementary Fig. 5a) exhibited a Lin⁻CD24⁺CD29^high population that is approximately four-fold larger (Fig. 4a, b). By contrast, the size of the CD24⁺CD29^high population in miR-31 KO mice was significantly smaller than in controls (Fig. 4c). A similar reduction was seen in K14-Cre;miR-31^fl/fl mammary epithelium-specific KO mice (Fig. 4d), indicating that miR-31 functions within the epithelium to regulate the size of the MaSC-containing basal cell compartment. A subpopulation of the CD24⁺CD29^high cells are known to express Lgr5 and exhibit enhanced regenerative capacity in mammary gland reconstitution assays[9], and the size of this Lgr5⁺ stem cell subpopulation was significantly reduced in miR-31 KO mammary glands (Fig. 4e and Supplementary Fig. 5b), which might account for the reduction of the CD24⁺CD29^high population in miR-31 KO mice.

To address whether miR-31 directly influences MaSC self-renewal, we performed *in vitro* colony formation assays using purified CD24⁺CD29^high cells from DTG mice. MiR-31 induction had no significant effect on the number of primary colonies that formed (Fig. 4f). However, the numbers of secondary, tertiary, and subsequent colonies were markedly increased by miR-31 induction, with overall colony formation potential showing no signs of exhaustion after six passages (Fig. 4f). Conversely, the addition of anti-miR-31 caused a remarkable reduction in colony formation upon serial passaging (Fig. 4f and Supplementary Fig. 5c). Colony sizes under control and Dox-treated conditions were relatively constant (Supplementary Fig. 5d). These results suggest that miR-31 promotes MaSC self-renewal.

When grown on feeder cell layers, basal progenitor cells form solid colonies, whereas luminal progenitor cells form acinar colonies with clear borders[29]. Sustained miR-31 expression preferentially increased the number of solid basal colonies at the cost of acinar colonies (Supplementary Fig. 5e), suggesting that miR-31 induction promotes the colony formation ability of basal progenitor cells while inhibiting their luminal differentiation or suppressing luminal cell colony formation.

To determine whether miR-31 positively regulates MaSC function in vivo, we performed limiting-dilution transplantation assays in immunodeficient mice using CD24⁺CD29^high cells isolated from either control or miR-31 KO mammary glands. The CD24⁺CD29^high cells from miR-31 KO mice displayed a significantly lower rate of successful engraftment and less extensive mammary outgrowth than the control cells (Fig. 4g). Collectively, these results provide strong evidence for the role for miR-31 in supporting MaSC self-renewal and mammary regeneration.

***Loss of miR-31 results in compromised tumorigenesis.*** To examine the role of miR-31 in mammary tumorigenesis, we bred the miR-31 KO allele into the MMTV-PyVT mouse model, which serves as the model for human breast adenocarcenomas[15]. The overall survival time of PyVT/KO mice was significantly longer than PyVT control mice (Supplementary Fig. 6a). Whole-mount analysis revealed that the development of primary mammary lesions was markedly suppressed in PyVT/KO mice at 12 and 16 weeks of ages (Fig. 5a). At 20 weeks of age, tumors in PyVT/KO mice were still much smaller than those in PyVT control mice (Fig. 5b). PyVT tumors displayed some malignant features (Fig. 5c), such as pleomorphic cell morphology and a moderate variation in nuclear morphology, size and shape, as early as at 12 weeks of age. These features were absent in PyVT/KO tumors, which morphologically appeared more like benign adenomas at this stage, characterized by relatively uniform cells with round to oval nuclei (Fig. 5c). In agreement with compromised tumor growth, the number of proliferating cells was much lower in PyVT/KO tumors than in PyVT controls (Fig. 5d and Supplementary Fig. 6b). Furthermore, transplantation assays with 4T1 mouse breast cancer cells revealed that inhibiting miR-31 with anti-miR-31 suppresses tumor growth in host mice (Supplementary Fig. 6c).

Gata3, the expression of which is strongly associated with ERα in breast cancer, is considered a prognostic marker for less aggressive breast cancer and associates with favorable outcome[45]. While Gata3- and ERα-positive cells were absent in PyVT tumors at 12 weeks of age, they were abundantly present in PyVT/KO tumors from the same stage (Fig. 5e, f). Consistently, the expression level of miR-31 also negatively correlated to Gata3 and ERα levels in human breast cancer (Supplementary Fig. 6d). K14⁺K8⁺ double positive cells have been reported to be tumor progenitor cells[46]. While PyVT tumors contained K14⁺K8⁺ cells at the leading, basal edge of the tumor (Fig. 5g), both tumors and mammary ducts from PyVT/KO mice lacked K14⁺ cells (Fig. 5g). The protein level of luminal marker K8 is much higher in PyVT/

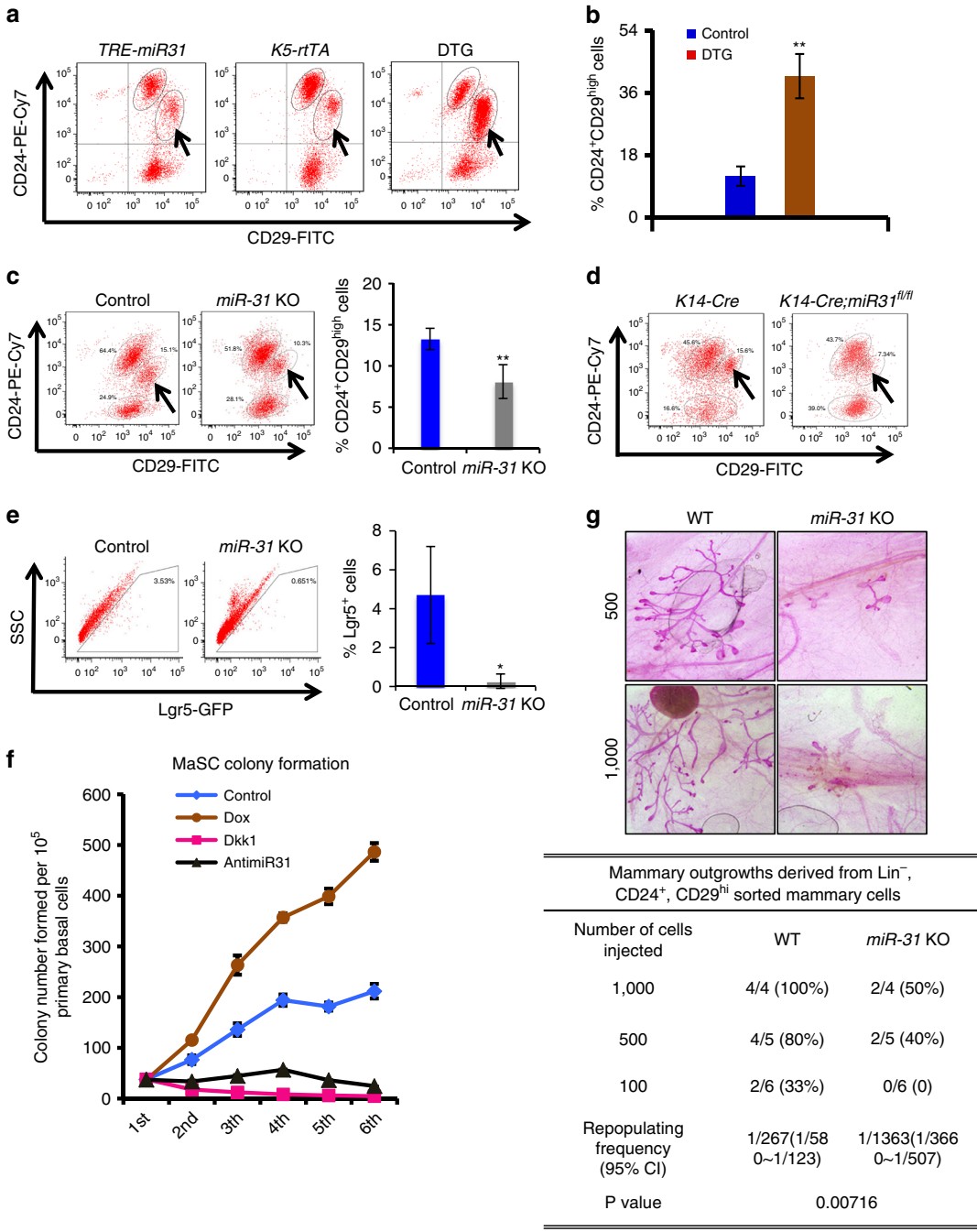

**Fig. 4** *MiR-31* induces mammary basal stem cell expansion. **a** Flow cytometry profiles of CD24-PE-Cy7 and CD29-FITC in cell suspensions from *TRE-miR31* ($n = 4$), *K5-rtTA* ($n = 4$) and DTG ($n = 4$) mammary glands at the same proestrus phase. Arrows indicate the CD24$^+$CD29$^{high}$ population. **b** Quantification of CD24$^+$CD29$^{high}$ cell populations in control (*TRE-miR31* or *K5-rtTA*, $n = 4$) and DTG ($n = 4$) mice. **c** Flow cytometry profiles of CD24-PE-Cy7 and CD29-FITC in cell suspensions from Control ($n = 3$) and *miR-31* KO ($n = 3$) mammary glands at the same proestrus phase at 12 weeks of age. Arrows indicate the CD24$^+$CD29$^{high}$ population. Quantification of CD24$^+$CD29$^{high}$ cells is shown to the right. **d** Flow cytometry profiles of CD24-PE-Cy7 and CD29-FITC in cell suspensions from *K14-Cre* and *K14-Cre;miR-31$^{fl/fl}$* mammary glands at the same proestrus phase at 10 weeks of age. Arrows indicate the CD24$^+$CD29$^{high}$ population. **e** GFP profiles of mammary epithelial cells from *Lgr5-eGFP-CreER* (Control, $n = 3$) and *Lgr5-eGFP-CreER;miR-31* KO (*miR-31* KO, $n = 3$) mice. Quantification of Lgr5-GFP$^+$ cells in control and *miR-31* KO mice. **f** Numbers of colonies formed by Control (scramble RNA), Dox, anti-*miR-31* and Dkk1 treated DTG mammary epithelial cells over six passages. $n = 4$ independent experiments. **g** Limiting dilution transplantation results. Top panel shows a representative whole mount analysis of repopulated mammary glands regenerated from 500 and 1000 CD24$^+$CD29$^{high}$ MaSC cells from Control and *miR-31* KO mice 7 weeks after transplantation. Bottom Table summarizes the transplantation results from 100 ($n = 6$), 500 ($n = 5$) and 1000 ($n = 4$) CD24$^+$CD29$^{high}$ MaSC cells. Data represented as mean ± S.D. $n \geq 3$. Two tailed unpaired *t*-test for **b**, **c**, **e** (*$P < 0.05$; **$P < 0.01$)

KO tumors, whereas basal marker K14 is greatly lower (Fig. 5f). These data suggest that loss of *miR-31* induces a basal-to-luminal cell fate change in mammary tumor cells. Thus, *miR-31* is required for maintaining a basal cell fate in both normal and malignant mammary epithelia. This notion is consistent with the presence of ample K14$^+$ cells in triple-negative tumors in which *miR-31* is more highly expressed as compared to hormone receptor-positive tumors (Supplementary Fig. 6e).

Cancer stem cells can be isolated from *PyVT* tumors using cell surface markers CD90 and CD24[47]. We found that the expression levels of *miR-31* are higher in the CD24⁺CD90⁺ cancer stem cells as compared to the CD24⁺CD90⁻ cells (Fig. 5h). We detected a significant reduction in the percentage of the CD24⁺CD90⁺ cancer stem cell population in *PyVT/KO* mammary tumors compared to *PyVT* tumors (Fig. 5i). Moreover, limiting-dilution transplantation assays revealed that *PyVT/KO* tumor cells show an approximately 13-fold decrease in tumor-initiating ability (Fig. 5j and Supplementary Fig. 6f), and

consequently tumor growth was compromised (Supplementary Fig. 6f–h). These data demonstrate that *miR-31* is required for maintaining mammary tumor stem cells.

Remarkably, *miR-31* deficiency in *PyVT* mice resulted in a higher metastasis-free survival rate (Fig. 6a). Consistently, we found a marked reduction of metastatic foci in *PyVT/KO* lungs, compared to *PyVT* controls (Fig. 6b). To probe the underlying mechanism, we examined the expression of known markers of epithelial-mesenchymal transition (EMT), which is arguably important for tumor invasion and metastasis[48–50]. The numbers

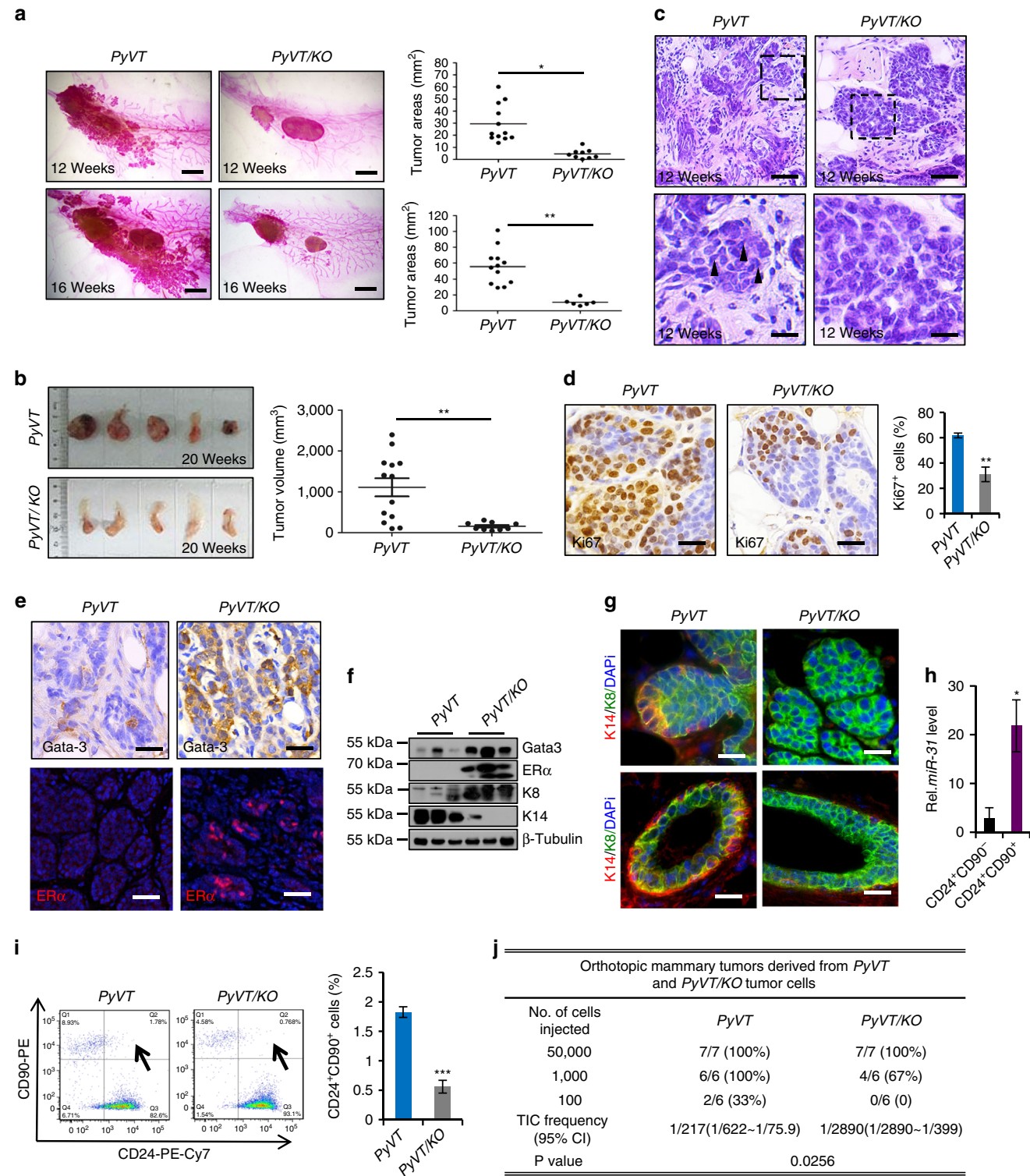

of vimentin[+] and Slug[+] cells were greatly reduced in *PyVT/KO* tumors (Fig. 6c, d). The RNA levels of other EMT regulators, *Twist1, Bmi1, Zeb1* and *Fzd3*, were also markedly decreased in *PyVT/KO* tumors (Fig. 6e). Moreover, the expression of P-cadherin, a metastable EMT marker of poor prognosis of invasive breast carcinomas[51], was reduced in *PyVT/KO* tumors and mammary ducts (Fig. 6f). However, the expression levels of α2- and β1-integrin, which are metastasis repressors in breast cancer[52], are remarkably elevated in *PyVT/KO* tumors (Fig. 6g, h). When cultured in matrigel of reconstituted basement membrane, *PyVT* tumor cells formed typical acinar structures with elongated protrusions, but such protrusions were blocked in *PyVT/KO* acini (Fig. 6i). In keeping with these in vivo/ex vivo findings, treatment of BT549 breast cancer cells with *miR-31* mimics elicited a morphological conversion from epithelial-like to mesenchymal-like state (Fig. 6j). Taken together, our findings reveal a critical involvement of *miR-31* in tumor growth and metastasis, likely through regulating multiple cellular processes including tumor stem cell self-renewal, basal-luminal differentiation and epithelial plasticity.

### MiR-31 activates Wnt/β-Catenin and represses TGFβ signaling.
To identify the signaling events that mediate this function, we examined the effect of *miR-31* alterations on the status of Wnt/β-catenin signaling, previously shown to be a key pathway that promotes MaSC self-renewal and stemness[5, 7, 53, 54]. The *Axin2-LacZ* reporter mice have been widely used as an in vivo reporter for Wnt pathway activity, as *Axin2* is a direct Wnt target in most cell types[7]. While *Axin2-LacZ* reporter activity was restricted to the single layer of basal cells in control mammary glands, it was expanded to multiple layers in DTG mammary glands and reduced in *miR-31* KO mammary glands (Fig. 7a). Increased nuclear localization of β-Catenin was seen in the mammary basal layer of DTG mice at 12 weeks of age, and this increase was more prominent during pregnancy (Fig. 7b). In contrast, expression of nuclear β-Catenin was considerably reduced in *miR-31* KO mammary tissues (Fig. 7c). These data, together with the finding of reduced number of cells that express Wnt downstream target Lgr5 in *miR-31* KO mammary glands (Fig. 4e), support a model where *miR-31* upregulates Wnt/β-catenin pathway activity in the mammary epithelium.

To ask whether *miR-31* impacts Wnt pathway activation in a cell-autonomous manner, we analyzed the effects of *miR-31* on Wnt signaling status/output in HC11 cells. *MiR-31* overexpression in this system induced prominent nuclear localization of β-Catenin (Fig. 7d), enhanced the expression of Wnt target LBH (Fig. 7e), and upregulated TOPflash Wnt reporter activity (Fig. 7f). In contrast, inhibiting *miR-31* with anti-*miR-31* led to decreased LBH expression (Fig. 7e) and reduced TOPflash reporter activity (Fig. 7f).

In interesting contrast to Wnt signaling, TGFβ signaling was found to be significantly elevated in *miR-31* KO mammary glands, as evident by increased level of active Smad2/3 (Fig. 7g), as well as upregulated expression of TGFβ pathway target genes *Cdkn2b* (encoding p15), *Cdkn1a* (encoding p21), *Cdkn1c* (encoding p57) and *Tgfbr1* (Fig. 7h). In addition, *miR-31* inhibition enhanced luciferase reporter expression driven by Smad-binding element CAGA, while *miR-31* mimics suppressed the expression of this TGFβ signaling reporter (Fig. 7i). Collectively, our results reveal opposite effects of *miR-31* on Wnt and TGFβ signaling in the mammary gland.

### MiR-31 targets Wnt signaling antagonists in mammary gland.
In order to identify direct molecular targets of *miR-31* that mediate its function in the mammary epithelia, we analyzed *miR-31*-binding sites in 3′-UTRs of transcripts encoding regulators of the Wnt signaling pathway. Putative *miR-31*-binding sites were found in several Wnt antagonists including *Axin1, Gsk3β* and *Dkk1* (Supplementary Fig. 7a). Indeed, a marked increase in mRNA and protein levels of *Axin1, Gsk3β* and *Dkk1* was detected in *miR-31* KO mammary glands, and conversely a significant reduction was observed in the levels of the same molecules in DTG mammary glands (Fig. 8a–c). In reporter luciferase assay, *miR-31* mimics significantly repressed luciferase activity containing *Axin1, Gsk3β* and *Dkk1* 3′-UTR elements, while 3′-UTR elements mutations abolished *miR-31* binding and this repressive effect (Fig. 8d and Supplementary Fig. 7b). Furthermore, RNA crosslinking, immunoprecipitation, and quantitative polymerase chain reaction with reverse transcription (qRT-PCR) (CLIP-PCR) assay revealed that *Dkk1, Axin1* and *Gsk3β* are enriched in the Ago2 antibody immunoprecipitates and that *miR-31* inhibitor suppresses these enrichments (Fig. 8e), suggesting that *miR-31* directly binds to these potential targets. Elevated *Axin1, Gsk3β* and *Dkk1* expression was also seen in *PyVT/KO* tumors as compared to *PyVT* tumors (Fig. 8f–h). Together, these findings underscore *Axin1, Gsk3β* and *Dkk1* as direct molecular targets of *miR-31*. Similar studies also implicate TGFβ pathway genes *Smad3* and *Smad4* as potential targets of *miR-31* (Supplementary Fig. 7 and Fig. 8).

Since the effects of *miR-31* on the mammary gland are most similar to those of Wnt signaling, we next asked whether inhibiting Wnt signaling is able to rescue the *miR-31* gain-of-function mammary phenotypes. Specifically, we crossed *miR-31* DTG mice with *TRE-Dkk1* transgenic mice to generate *K5-rtTA/TRE-miR31/TRE-Dkk1* (DTG/Dkk1) triple transgenic mice. Mammary glands were harvested from these mice at 12 weeks of age and the altered expression of *miR-31* and/or *Dkk1* was confirmed (Supplementary Fig. 8a, b). Distinct from DTG or *Dkk1* transgenic mammary glands, DTG/*Dkk1* triple transgenic

**Fig. 5** Loss of *miR-31* results in compromised mammary tumor progression and reduces number of cancer stem cells. **a** Representative whole-mount staining of *MMTV-PyVT* (*PyVT*) and *MMTV-PyVT/miR-31KO* (*PyVT/KO*) mammary glands at 12 and 16 weeks of age. Quantification of *PyVT* tumor areas at 12 weeks (*PyVT*: $n = 12$; *PyVT/KO*: $n = 9$) and 16 weeks (*PyVT*: $n = 12$; *PyVT/KO*: $n = 6$) of age. Scale bar, 2.5 mm. **b** Gross images of mammary tumors from *PyVT* and *PyVT/KO* mice at 20 weeks of age. Quantification of mammary tumor volume from *PyVT* ($n = 13$) and *PyVT/KO* ($n = 9$) mice at 20 weeks of age. **c** H&E of *PyVT* and *PyVT/KO* tumors at 12 weeks of age. Higher magnification images indicated by dashed boxes shown in the lower panels. Arrowheads, nuclei with diverse shape. Scale bar, top panels, 50 μm; lower panels, 16 μm. **d** Immunohistochemistry for Ki67 and quantification of Ki67[+] cells in *PyVT* and *PyVT/KO* mammary tumors at 12 weeks of age. Scale bar, 25 μm. **e** Immunostaining for Gata-3 and ERα in *PyVT* and *PyVT/KO* mammary tumors at 12 weeks of age. Scale bar, 25 μm. **f** Western blotting for Gata-3, ERα, K8 and K14 in *PyVT* and *PyVT/KO* tumors at 12 weeks of age. **g** Immunofluorescence for K14 and K8 at the leading edges of tumors and mammary ducts from *PyVT* and *PyVT/KO* mice. Scale bar, 25 μm. **h** qRT-PCR for *miR-31* in CD24[+]CD90[-] and CD24[+]CD90[+] cells from *PyVT* tumors. $n = 3$ biological replicates. **i** Profiles of CD90-PE and CD24-PE-Cy7 in cell suspensions from *PyVT* and *PyVT/KO* tumors. Arrows, CD24[+]CD90[+] cells. Quantification of CD24[+]CD90[+] cells in *PyVT* ($n = 4$) and *PyVT/KO* ($n = 4$) tumors. **j** Limiting dilution transplanted assay showing tumor-initiating cell frequencies (with confidence intervals) $\chi^2$-values and associated probabilities. $P = 0.0256$. Data represented as mean ± S.E.M. for **a**, **b**; Data represented as mean ± S.D. for **d**, **h**, **i**. Sample size: *PyVT* ($n = 3$) and *PyVT/KO* ($n = 3$) for **c**–**g**. Two tailed unpaired *t*-test for **d**, **h**, **i** (**P* < 0.05; ***P* < 0.01; ****P* < 0.001)

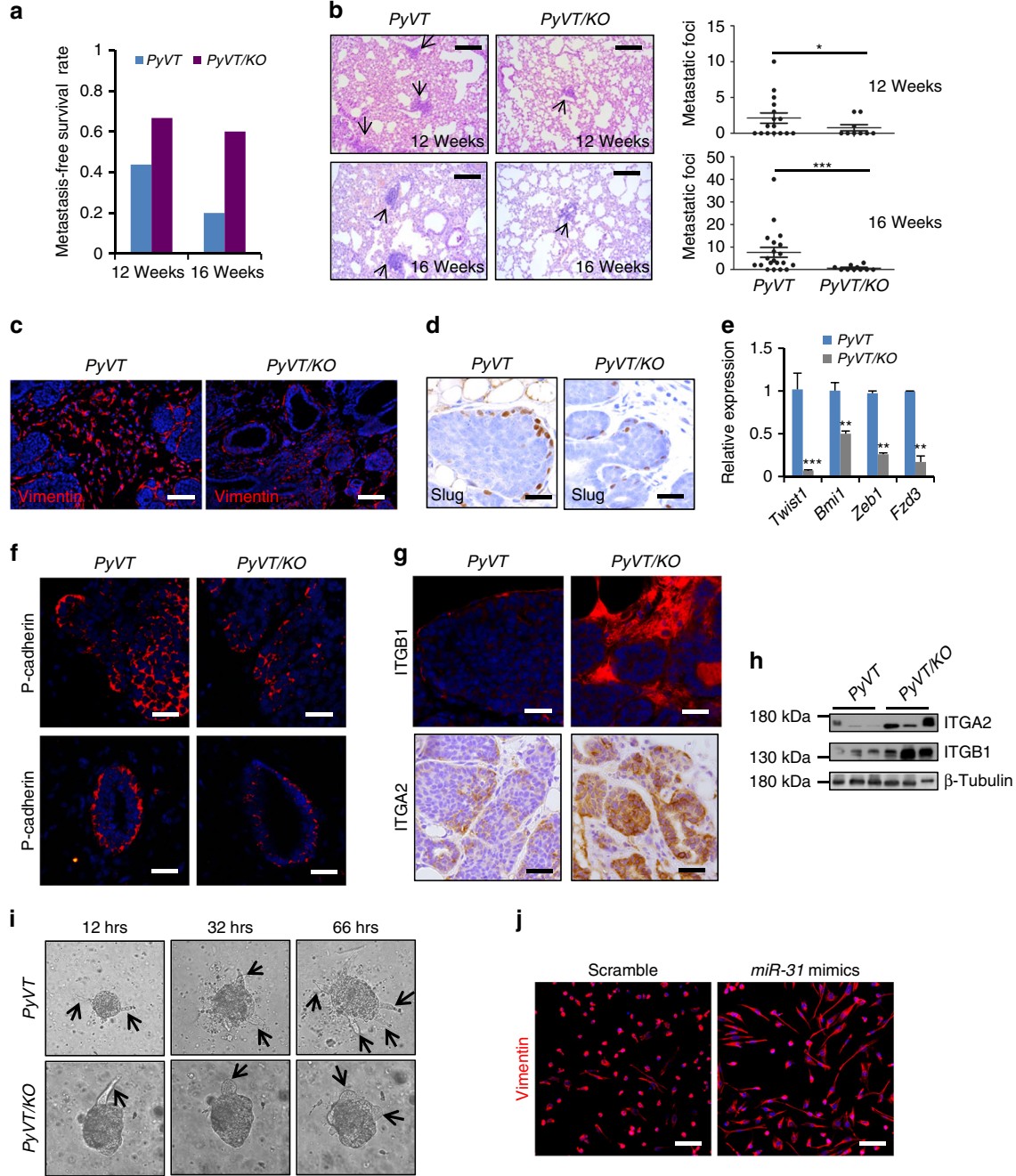

**Fig. 6** Loss of *miR-31* represses metastasis to the lung. **a** Quantification of metastasis-free survival rates of *PyVT* and *PyVT/KO* mice at 12 and 16 weeks of age. 12 wks: *PyVT* (*n* = 16), KO/*PyVT* (*n* = 9); 16 wks: *PyVT* (*n* = 20), KO/*PyVT* (*n* = 10). **b** Representative histology of lungs from *PyVT* and *PyVT/KO* mice at 12 and 16 weeks of age. Arrows point to metastatic foci in the lungs. Quantification of metastatic tumors in the lungs. 12 wks: *PyVT* (*n* = 16), KO/*PyVT* (*n* = 9); 16 wks: *PyVT* (*n* = 20), KO/*PyVT* (*n* = 10). Scale bar, 100 μm. **c** Immunofluorescence for Vimentin in *PyVT* and *PyVT/KO* tumors at 12 weeks of age. Scale bar, 50 μm. **d** Immunohistochemistry for Slug in *PyVT* and *PyVT/KO* tumors at 12 weeks of age. Scale bar, 25 μm. **e** qRT-PCR analysis for *Twist1*, *Bmi1*, *Zeb1* and *Fzd3* in *PyVT* and *PyVT/KO* tumors at 12 weeks of age. *n* = 3 biological replicates. **f** Immunofluorescence for P-cadherin in tumors (top panels) and normal mammary ducts (bottom panels) from *PyVT* and *PyVT/KO* mice at 12 weeks of age. Scale bar, 25 μm. **g** Immunofluorescence for β1-integrin (ITGB1) and IHC for α2-integrin (ITGA2) in *PyVT* and *PyVT/KO* tumors at 18 weeks of age. Scale bar, 25 μm. **h** Western blotting for ITGA2 and ITGB1 in *PyVT* and *PyVT/KO* tumors at 18 weeks of age. GAPDH was used as a loading control. **i** Morphology of *PyVT* and *PyVT/KO* tumor cells into three-dimensional matrigel culture at indicated time points. Arrows point to protrusions. **j** Immunofluorescence for Vimentin in BT549 breast cancer cells treated with Scramble RNA and *miR-31* mimics. Scale bar, 50 μm. *n* = 3 technical replicates. Data represented as mean ± S.D. Sample size: *PyVT* (*n* = 3) and *PyVT/KO* (*n* = 3) for **c–i**. Two tailed unpaired *t*-test for **e** (*$P < 0.05$; **$P < 0.01$; ***$P < 0.001$)

glands exhibited structural features similar to those of controls, including thin ducts, underdeveloped tertiary branches, but otherwise normal ductal elongation and branching (Supplementary Fig. 8c). DTG/*Dkk1* mammary ducts had fewer cell layers than DTG ducts (Supplementary Fig. 8c). Furthermore, the

number of CD24⁺CD29^high cells in DTG/*Dkk1* mice is much lower than that of DTG mice but higher than WT mice (Fig. 8i). Consistently, administration of recombinant Dkk1 protein suppressed colony formation in vitro (Fig. 4f). Taken together, these findings indicate that inhibiting Wnt signaling by

introducing Dkk1 can partially rescue the DTG mammary morphology, histology and MaSC expansion defects in vivo. Our data support a model where *miR-31* functions in the mammary gland at least in part through suppressing Wnt signaling antagonists, thereby maintaining optimal Wnt pathway activation that is crucial for MaSC/basal cell self-renewal.

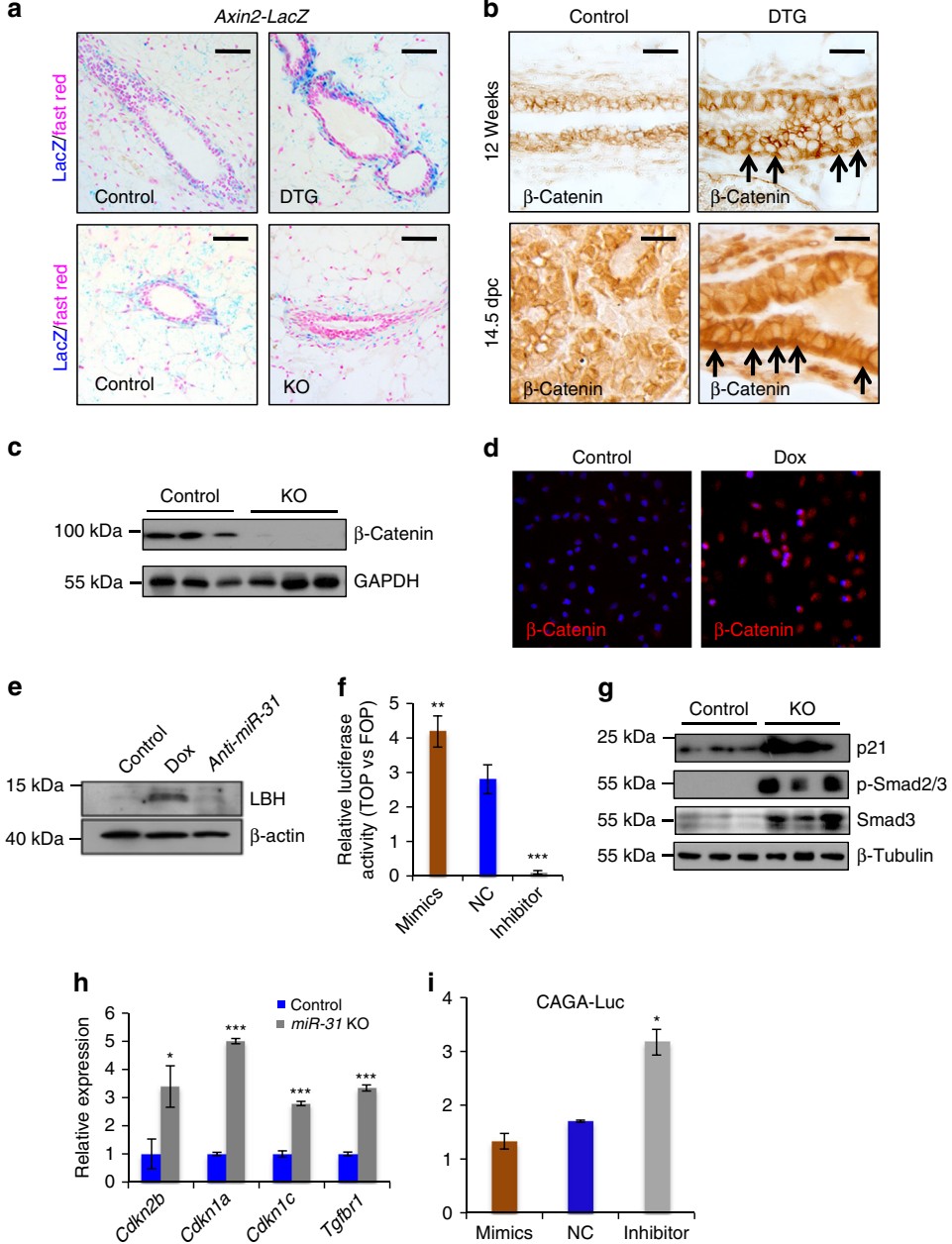

**Fig. 7** *MiR-31* activates Wnt and represses TGFβ signaling pathways. **a** Wnt signals were evaluated by *Axin2-LacZ* reporter activity in mammary ducts from Control ($n = 3$) and DTG ($n = 3$) mice, as well as Control ($n = 3$) and *miR-31* KO ($n = 3$) mice. Blue, LacZ signals. Scale bar, 100 μm. **b** Immunohistochemistry for β-Catenin in Control ($n = 3$) and DTG ($n = 3$) mammary glands at 12 weeks of age and 14.5 d.p.c. Arrows, nuclear localized β-Catenin. Scale bar, 25 μm. **c** Western blotting for β-Catenin in Control ($n = 3$) and *miR-31* KO ($n = 3$) mice. GAPDH was used as a loading control. **d** Immunofluorescence for β-Catenin in HC11 mouse mammary epithelial cells in the presence of Dox (*miR-31* overexpression). Control, without Dox. Red, β-Catenin; Blue, DAPI. $n = 3$ technical replicates. **e** Western blotting for Wnt target LBH in *miR-31* overexpressing (Dox) or *miR-31* inhibited (*anti-miR-31*) HC11 mammary epithelial cells. The Control HC11 cells were treated with Scramble RNA. $n = 3$ technical replicates. **f** TOP/FOPflash luciferase assays demonstrate that Dox induced *miR-31* overexpression activates Wnt reporter gene expression and that *miR-31* inhibitor (*anti-miR-31*) markedly represses Wnt reporter gene expression. Scramble RNA was used as negative control (NC). $n = 3$ technical replicates. **g** Western blotting for Smad3, p-Smad2/3 and p21 in mammary epithelial cells of Control ($n = 3$) and *miR-31* KO ($n = 3$) mice at 10 weeks of age. **h** qRT-PCR analysis for TGFβ components and downstream target genes, *Cdkn2b* (encoding p15), *Cdkn1a* (encoding p21), *Cdkn1c* (encoding p57) and *Tgfbr1* in mammary epithelial cells of Control ($n = 3$) and *miR-31* KO ($n = 3$) mice at 10 weeks of age. **i** HC11 mammary epithelial cells were transfected with CAGA luciferase reporter vector, combined with scrambled RNA (negative control, NC), *miR-31* mimics or *miR-31* inhibitor (*anti-miR-31*) for 24 h and then harvested for luciferase activity determination. $n = 3$ biological replicates. Data represented as mean ± S.D. $n = 3$. Two tailed unpaired *t*-test for **f**, **h**, **i** (*$P < 0.05$; **$P < 0.01$; ***$P < 0.001$)

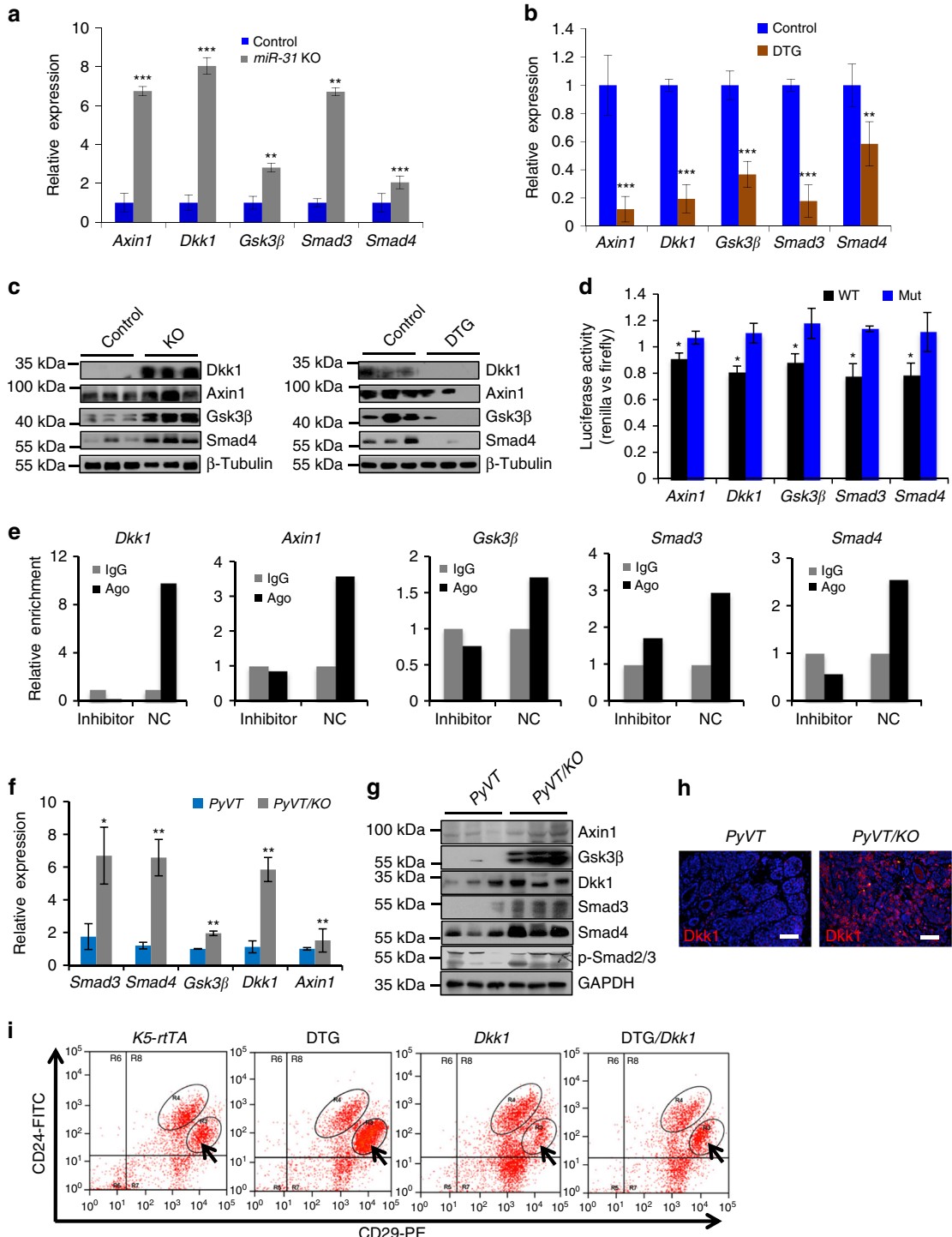

**Fig. 8** Identification of *miR-31* direct targets. **a**, **b** qRT-PCR analysis for *Axin1*, *Dkk1*, *Gsk3β*, *Smad3* and *Smad4* in Control ($n = 3$) and *miR-31* KO ($n = 3$) **a**, as well as Control ($n = 3$) and DTG ($n = 3$) **b** mammary epithelium. **c** Western blotting for Dkk1, Axin1, Gsk3β and Smad4 in Control ($n = 3$) and *miR-31* KO ($n = 3$), as well as Control ($n = 3$) and DTG ($n = 3$) mammary epithelium. β-Tubulin was used as a loading control. **d** Ratio of luciferase activity of *miR-31* mimics vs. scrambled RNA in wild type (WT) and mutant (Mut) 3′-UTR constructs based on 3 independent experiments. **e** RNA crosslinking, immunoprecipitation, and qRT-PCR (CLIP-PCR) assay for *Dkk1*, *Axin1*, *Gsk3β*, *Smad3* and *Smad4* upon Ago2 antibody immunoprecipitates in response to *miR-31* inhibitor and scramble RNA (NC). IgG was used as a negative control. **f** qRT-PCR analysis for *Smad3*, *Smad4*, *Axin1*, *Dkk1* and *Gsk3β* in *PyVT* ($n = 3$) and *PyVT/KO* ($n = 3$) tumors. **g** Western blotting for Dkk1, Axin1, Gsk3β, Smad4 and p-Smad2/3 in *PyVT* ($n = 3$) and *PyVT/KO* ($n = 3$) tumors at 12 weeks of age. β-Tubulin was used as a loading control. **h** Immunofluoresence for Dkk1 in *PyVT* ($n = 3$) and *PyVT/KO* ($n = 3$) tumors at 12 weeks of age. Scale bar, 50 μm. **i** Flow cytometry profiles of CD24-FITC and CD29-PE in cell suspensions of mammary epithelium from *K5-rtTA* ($n = 3$), DTG ($n = 3$), *Dkk1* ($n = 3$) and DTG/*Dkk1* ($n = 3$) mice. Arrows indicated CD24$^+$CD29$^{high}$ cell population. $n = 3$ biological replicates. Data represented as mean ± S.D. $n = 3$. Two tailed unpaired *t*-test for **a**, **b**, **d**, **f** (*$P < 0.05$; **$P < 0.01$; ***$P < 0.001$)

## Discussion

In this work, we demonstrate that *miR-31* plays an important role in MaSC self-renewal and tumorigenesis by regulating Wnt pathway activation. Our work adds *miR-31* to a short list of microRNAs, including *miR-22* and *miR-205*, which regulate stem/progenitor cells in the mammary lineage. Ectopic *miR-22* overexpression results in increased mammary ductal side-branching accompanied by an apparent expansion of MaSCs[32]. Both overexpression and knockdown of *miR-205* have been shown to cause perturbations of stem/progenitor-like populations[28, 33], yet in vivo evidence is still lacking. To our knowledge, this study is the first to demonstrate the in vivo, physiological role of a specific microRNA in regulating MaSCs and mammary gland development using a loss-of-function mouse model (Supplementary Fig. 9).

The major cellular consequences of *miR-31* loss are reduction of MaSCs, impaired cell proliferation, and precocious alveolar differentiation. Our findings suggest that reduced Wnt signaling is at least in part responsible for these effects. It has been shown that restricting Wnt activity leads to reduction in the frequency/self-renewal of MaSCs and increased luminal differentiation[54–56]. Wnt targets such as *Axin2* and *Lgr5* have been used to identify distinct Wnt-responsive MaSC subsets[7, 9]. We found that both Axin2-LacZ[+] and Lgr5[+] cells are reduced in *miR-31* KO mammary glands, which could account for the reduced number of MaSCs in KO mammary glands. On the other hand, the mammary phenotype of DTG mice mimicked hyperplasia and MaSC-enriched cell expansion resulting from ectopic expression of Wnt1 or stabilized β-Catenin in transgenic mice[5, 57]. In serial passages, *miR-31* induction resulted in increased numbers of colonies, which resemble the effect of Wnt3a on mammary stem cells[7]. Moreover, *miR-31* overexpression induced prominent nuclear localization of β-Catenin, enhanced the expression of *Axin2*, and upregulated TOPflash Wnt reporter activity, while *miR-31* inhibition exerted opposite effects. Collectively, these data support the importance of *miR-31* in activating the Wnt/β-Catenin signaling pathway. We surmise that *miR-31* does so primarily in the mammary basal layer to sustain MaSC/basal cell self-renewal/proliferation and to suppress their differentiation into luminal/alveolar cells.

Our results also suggest *miR-31* as a pro-oncogenic microRNA in the mammary tissue. Our findings that loss of *miR-31* in *MMTV-PyVT* mice (which develop multifocal mammary adenocarcinomas and metastatic lesions in the lung[58]) leads to compromised tumor growth, reduced number of cancer stem cells, decreased tumor-initiating ability and impaired metastasis to the lung, point to an in vivo pro-oncogenic role for *miR-31* in breast cancer. Previous studies have reported an anti-metastatic role for *miR-31* in breast cancer[59–62]. However, this conclusion was based on in vitro data as well as an inverse correlation between *miR-31* expression and metastatic potential of breast cancer subtypes and cell lines[59–62]. As such, our study represents the first report of *miR-31*'s physiological role in mammary tumorigenesis. K14[+]K18[+] cells are thought to be the progenitor cells in breast tumorigenesis[46], and K14[+] cells are capable to collectively invade in human breast cancer and mouse models of breast cancer[63]. Moreover, Gata3 and ERα have been identified as clinical markers for the less aggressive luminal A-type tumors[45]. Loss of *miR-31* leads to a remarkable reduction of K14[+]K18[+] and K14[+] cells as well as a significant increase in Gata3- and ERα-positive cells in *PyVT* tumors. These data suggest that loss of *miR-31* promotes an aberrant basal-to-luminal cell fate transition within the tumors, and offer a possible cellular mechanism for reduced tumor formation and metastasis (Supplementary Fig. 9). Our work, together with the previous finding that *miR-31* also functions as an oncomir in colon cancer and lung cancer[64, 65],

highlights *miR-31* as a potential therapeutic target for cancers of multiple tissues.

## Methods

**Mouse strains.** To generate *TRE-miR-31* transgenic mice, the *mmu-miR-31* sequence was amplified using the following primers: Forward 5′-CTCGGATCCTGTGCATAA CTGCCTTCA-3′ (BamHI site is added), and Reverse 5′-CACAAGCTTGAAGT CAGGGCGAGACAGAC-3′ (HindIII site is added), from mouse tail DNA, and was inserted into *pTRE2* vector (Clontech) to generate a *pTRE2-miR-31* construct. *TRE-miR31* transgenic mice were produced using standard protocols. *K5-rtTA* mice[66] were crossed with *TRE-miR31* mice to create *K5-rtTA/TRE-miR31* double transgenic mice (DTG). *Lgr5-EGFP-CreERT* mice were purchased from the Jackson Laboratory (#008875). Constitutive *miR-31* KO mice were generated using CRISPR/Cas9 RNA-guided nucleases at the Nanjing Animal Center, and 402 bp DNA fragment containing *miR-31* was deleted in the null allele. The conditional *miR-31* KO allele was generated at the Shanghai Model Animal Center, the first exon (14,806-15,522) of *miR-31* was targeted with flanking LoxP sites resulting in the 2 LoxP locus. *K14-Cre* transgenic mice were obtained from Jackson Laboratory (stock no.004782). The genetic background of miR-31 KO, K14-Cre, cKO and TRE-miR31 transgenic mice is C57BL/6J and that of MMTV-PyVT mice is Friend Virus B-type (FVB). All females were used for analysis in this study. The ages of mice in the experiments have been described in the Text. To quantify the tumor areas, all lesions in mammary whole-mount section were measured using Photoshop software. At least three control and three DTG, or three control and three KO littermates were used to analyze the mammary phenotypes. For xenograft assay, at least six mice each group were used to be transplanted with breast cancer cells.

**Ethics.** All mouse experiment procedures and protocols were evaluated and authorized by the Regulations of Beijing Laboratory Animal Management and were strictly in accordance with the guideline under the Institutional Animal Care and Use Committee of China Agricultural University(approval number, SKLAB-2011-04-03).

**Cell sorting and transplantation assay.** Mouse mammary glands and *PyVT* tumors were prepared according to the manufacturer protocol (Stem Cell Technologies, Vancouver, Canada). The minced mammary gland and *PyVT* tumors tissues were dissected from age-matched virgin or pregnant mice in EpiCult-B with 5% fetal bovine serum (FBS), 300 U/ml collagenase and 100 U/ml hyaluronidase for 3–6 hrs at 37 °C. After lysis of the red blood cells with NH$_4$Cl, a single cell suspension was obtained by sequential dissociation of the fragments in 0.25% trypsin for 1–2 min and 5 mg/ml dispase plus 0.1 mg/ml DNase I (DNase; Sigma) for 2 min with gentle pipetting, followed by filtration through 40-μm cell strainer (BD Falcon). The following antibodies were used: CD24-rphycoerythrin-Cy7 (560536, BD), CD29-fluorescein isothiocyanate (561796, BD), CD45-allophycocyanin (APC, 559864, BD), mouse CD31-APC (551262, BD), mouse TER-119 APC (557909, BD), CD90.2-PE (553005, BD) and Fixable Viability Dye eFluor 450 (eBioscience, 65-0863-14). Antibody incubation was performed on ice for 15 min in HBSS with 2% fetal bovine serum. Cell sorting was performed using a FACSAria SORP (BD). Sorted cells were re-suspended in 50% Matrigel, PBS with 20% fetal bovine serum, and injected in 10 μl volumes into the cleared fat pads of 3-week-old femal NOD-SCID mice. Reconstituted mammary glands were examined after 6–8 weeks post injection. Outgrowths were detected under dissection microscope (Leica) after Carmine staining. For limiting dilution analyses, the repopulating frequency after was calculated using the Extreme Limiting Dilution Analysis Program.

**In vitro colony formation assays.** Colony formation assays were performed according to published protocols[7] with modifications. We performed fluorescence-activated cell sorting (FACS) assay to sort CD24[+]CD29[hi] cells from DTG mammary glands without Dox treatment. The sorted CD24[+]CD29[hi] cells were equally divided into three aliquots, each containing 10[5] cells. One aliquot was electro-transfected with 4 μg of *miR-31* inhibitor, anti-miR-31 (5′-CAGCUAUGCCAG-CAUCUUGCCU-3′) (Shanghai GenePharma Co., Ltd). The other aliquot was electrotransfected with 4 μg scrambled RNA (5′-CAGUACUUUUGUGUAGUA-CAA-3′). Next, each aliquot was plated onto 0.4 μm Culture Plate Inserts (Millipore Cor., Billerica, USA). The cells were incubated in Complete EpiCult-B Medium containing EpiCult-B proliferation supplements, 10 ng/ml recombinant human Epidermal Growth Factor (rhEGF; Stem Cell Technologies), 10 ng/ml recombinant human Basic Fibroblast Growth Factor (rh bFGF; Stem Cell Technologies), 4 μg/ml (0.0004%) Heparin (Stem Cell Technologies), 100 U/mL Penicillin and 100 μg/ml of Streptomycin (Sigma), and 10% FCS (Hyclone). After 24 h, the culture medium was changed to serum-free Complete EpiCult-B Medium containing cytokines. The culture medium was changed every 24 h thereafter. Colony numbers and size were scored after 8 days in culture (first passage). After scoring, colonies were digested in the corresponding wells by incubation in 1–2 ml 0.25% trypsin⁻EDTA for 3–5 min at 37 °C. Colonies released from culture plate inserts were pelleted, followed by gentle pipetting to obtain single cells. The anti-miR31-treated cells were eletrotransfected again with 4 μg anti-miR-31 RNA. The

other two aliquots were eletrotransfected with scrambled RNA. The three aliquots were replated in Culture Plate Inserts. One of the two aliquots eletrotransfected with scrambled RNA was treated with 200 ng/μl Dox to induce *miR-31* over-expression. The remaining aliquot was cultured without Dox. 8 days after culture, colony numbers and size were scored again (second passage). The following 3–6 passages were repeated as above.

Colony formation assays on feeder layers were performed as described by Shackleton et al.[5]. We sorted Lin− mammary epithelial cells from transgenic mammary glands without Dox treatment, and plated approximately 10,000 mammary epithelial cells onto irradiated NIH3T3 feeders. Solid and acinar colonies were observed and compared with or without Dox treatment for 6 days on feeders.

**Cell culture and stable transfection using a Tet-on system.** HC11 mouse mammary epithelial cells were cultured in RPMI-1640 medium (R8758, Sigma) supplemented with 10% FBS (Hyclone), 10 ng/ml epithelial growth factor (EGF), 5 μg/ml insulin, and grown to 80% confluence, then changed into Opti-MEM medium without serum and antibiotics so that cells would be 90–95% confluent at the time of transfection. 4 μg of pTet-On plasmid (Clontech) were transfected using Lipofectamine™ 2000 according to the manufacturer's instructions. Positive clones were selected following treatment with 800 μg/ml G418 for 2 weeks. Clones were seeded into 24-well plates with RPMI-1640 medium and genotyped by PCR. Positive clones were co-transfected with *TRE-miR31* plasmid and selection vector pTK-Hyg (Clontech) at a molar ratio of 20:1, and selected by addition of 600 μg/ml Hygromycin for 2 weeks. Surviving clones were PCR genotyped. No mycoplasma contamination was detected in any of the cultures using a mycoplasma detection kit.

**In vitro lactogenic differentiation assay.** HC11 mouse mammary epithelial cells were cultured in RPMI-1640 growth medium supplemented with 10 ng/ml EGF, 5 μg/ml insulin and 10% FBS. For induction of lactogenic differentiation, lactogenic hormones (1 μg/ml dexamethasone, 5 μg/ml insulin and 5 μg/ml prolactin (DIP)) were added to RPMI-1640 medium for 6 days. The cells were treated with 200 ng/ml Doxycycline (Dox), *miR-31* inhibitor (*Anti-miR-31*, GenePharma Co., Ltd), an artificial RNA sequence (5′-CAGCUAUGCCAGCAUCUUGCCU-3′), or scrambled (5′-CAGUACUUUUGUGUAGUACAA-3′). Three independent replicates were performed for each experiment.

**Histology and immunostaining.** Mammary glands were fixed in 4% PFA, paraffin-embedded and 5-μm sections were stained with hematoxylin and eosin (H&E). For immunohistochemistry staining, antigen-retrieval was performed by heating slides to 95 °C for 10 min in 0.01 M citrate buffer (pH 6) in a microwave oven. The sections were then immunostained by the ABC peroxidase method (Vector Laboratories) with diaminobenzidine as the enzyme substrate and hematoxylin as a counterstain. For detection of nuclear β-Catenin, antibody clone 15B8 (Sigma; 1:1,000) was used in combination with the MOM kit (Vector Laboratories). For immunofluorescence staining, paraffin sections were microwave pretreated, and incubated with primary antibodies, then incubated with secondary antibodies (invitrogen) and counterstained with DAPI in mounting media. The following antibodies were used: K14 (1:400, ab7800, Abcam), K18 (1:400, ab668, Abcam), ERα (1:500, sc-542, Santa Cruz), PR (Cell Signaling Technology), Cyclin D1 (1:100, ab134175, Abcam), c-Myc (1:200, ab32072, Abcam), Ki67 (1:800, sp6, Thermo Fisher), β-catenin (1:400, 8480, Cell Signaling Technology), p-Stat5 (1:500, 9359, Cell Signaling Technology), P-cadherin (1:200, sc-7893, Santa Cruz), p63 (1:500, ab124762, Abcam), β-casein (1:400, sc-166530, Santa Cruz), GFP (1:500, ab290, Abcam), Rankl (1:500, ab45039, Abcam), SMA (1:500, A5228, Sigma), k8 (1:500, ab59400, Abcam), Gata3 (1:800, ab199428, Abcam), Prlr (1:400, sc-20992, Santa Cruz), Slug (1:500, ab27568, Abcam), Vementin (1:400, 5741, Cell Signaling Technology), ITGA2 (1:800, ab133557, Abcam), ITGB1 (1:800, ab52971, Abcam), Dkk1 (1:500, ab61275, Abcam).

**Whole mount staining.** Inguinal mammary glands were spread on glass slides, fixed in Carnoy's fixative (6:3:1, 100% ethanol: chloroform: glacial acetic acid) for 2–4 h at room temperature, washed in 70% ethanol for 15 min, rinsed through graded alcohol followed by distilled water for 5 min, then stained in carmine alum overnight, washed in 70, 95 and 100% ethanol for 15 min each, cleared in xylene and mounted with Permount. After photographic documentation, tissues were immersed in xylene to remove mounting medium, transferred to 1:1 xylene: paraffin (60 °C) and embedded in paraffin for sectioning and histological analysis.

**Western blotting.** Mammary epithelial cell lysates were prepared in lysis buffer (150 mM NaCl, 50 mM Tris-Cl, 1% Triton X-100, 0.1% SDS, 0.5% sodium deoxycholate, 0.02% sodium azide, 1 mM sodium vanadate, protease inhibitors (10 μg/ml leupeptin, 10 μg/ml aprotinin, and 1 mM phenylmethylsulfonyl fluoride)). Protein concentrations were determined using a Bio-Rad protein assay. 20 μg protein was loaded in 4–12% sodium dodecyl sulphate–polyacrylamide gel electrophoresis (Invitrogen), and transferred to nitrocellulose. Membranes were blocked in 5% BSA for 1 h, incubated with the primary antibody overnight at 4 °C,

washed, and incubated with the appropriate horseradish-conjugated secondary antibody (1:5,000) for 1 h at room temperature. An enhanced chemiluminescent (ECL) kit was used to visualize the signal (Pierce). The following antibodies and dilutions were used: Dkk1 (1:500, ab61275, Abcam), Axin1 (1:400, sc-14029, Santa Cruz), LBH (1:400, sc-161791, Santa Cruz), β-catenin (1:400, 8480, Cell Signaling Technology), and β-actin (1:400, sc-1616, Santa Cruz), Gsk3β (1:5,000, ab32391, Cell Signaling Technology), p-Smad2/3 (1:1000, 8828, Cell Signaling Technology), Smad4 (1:400, sc-7966, Santa Cruz), Rankl (1:500, ab45039, Abcam), p-p65 (1:1,000, 3033, Cell Signaling Technology), p65 (1:1,000, 8242, Cell Signaling Technology), GAPDH (1:2000, 2118, Cell Signaling Technology), p-ikkα (1:1,000, ab38515, Abcam), p-AKT (1:1000, 9614, Cell Signaling Technology), Gata3 (1:2,000, ab199428, Abcam), Prlr (1:500, sc-20992, Santa Cruz), Cyclin D1 (1:1,000, ab134175, Abcam), c-Myc (1:1,000, ab32072, Abcam), p-Stat5 (1:1000, 9359, Cell Signaling Technology), β-Tubulin (1:4,000, ab6046, Abcam), β-casein (1:500, sc-166530, Santa Cruz), ERα (1:1,000, sc-542, Santa Cruz), k8 (1:2,000, ab59400, Abcam), K14 (1:2,000, ab7800, Abcam), ITGA2 (1:2,000, ab133557, Abcam), ITGB1 (1:2,000, ab52971, Abcam), p21(1:1,000, ab109520, Abcam), smad3 (1:1000, ab40854, Abcam), P-cadherin (1:500, sc-7893, Santa Cruz). Uncropped blots for western blot analysis are shown in Supplementary Fig. 10.

**Quantitative RT-PCR analysis.** Total RNA was isolated from mammary epithelial cells using the mirVana™ RNA Isolation kit following manufacturer's instructions (Ambion). Complementary DNA was prepared using the MMLV cDNA synthesis kit (Promega). qRT-PCR was performed using the SYBR-green detection system (Roche). For microRNA expression, mature *miR-31* was quantified using TaqMan microRNA assays according to the manufacturer's instructions. U6 snRNA was used as an internal control (Applied Biosystems). qRT-PCR primers as follows:

*Ccnd1*-forward: 5′-GCAGGAGAGGAAGTTGTTGG-3′;
*Ccnd1*-reverse: 5′-AGACCTTTGTGGCCCTCTGT-3′.
*c-Myc*-forward: 5′-TCCTGTACCTCGTCCGATTC-3′;
*c-Myc*-reverse: 5′-GGTTTGCCTCTTCTCCACAG-3′.
*Axin-1*-forward: 5′-CCCCCATACAGGATCCGTAAG-3′;
*Axin-1*-reverse: 5′-AGAGGTACCCGCCCATTGA-3′.
*Dkk1*-forward: 5′-TACAATGATGGCTCTCTGCAGCCT-3′;
*Dkk1*-reverse: 5′-GCAGGTTCTTGATCGCGTTGGAAT-3′.
*Gsk3β*-forward: 5′-CCAACAAGGGAGCAAATTAGAGA-3′;
*Gsk3β*-reverse: 5′-GGTCCCGCAATTCATCGAAA-3′.
*Smad3*-forward: 5′-ACAGGCGGCAGTAGATAACG-3′;
*Smad3*-reverse: 5′-AACGTGAACACCAAGTGCAT-3′.
*Smad4*⁻-forward: 5′-GGCTGTCCTTCAAAGTCGTG-3′;
*Smad4*-reverse: 5′-GGTTGTCTCACCTGGAATTGA-3′.
*Cdkn2b* -forward: 5′-GCCCAATCCAGGTCATGATG-3′;
*Cdkn2b* -reverse: 5′-TCACACACATCCAGCCGC-3′.
*Cdkn1a* -forward: 5′-ATCACCAGGATTGGACATGG-3′;
*Cdkn1a* -reverse: 5′-CGGTGTCAGAGTCTAGGGGA-3′.
*Cdkn1c* -forward: 5′-GTTCTCCTGCGCAGTTCTCT-3′;
*Cdkn1c* -reverse: 5′-GTTCTCCTGCGCAGTTCTCT-3′.
*Tgfbr1* -forward: 5′-CAACCCAGGTCCTTCCTAAA-3′;
*Tgfbr1*-reverse: 5′-GGAGGAGCCCTGGATACCAAC-3′.
*Smad7*-forward: 5′-CTTCTCCTCCCAGTATGCCA-3′;
*Smad7*-reverse: 5′-GAACGAATTATCTGGCCCCT-3′.
*GAPDH* -forward: 5′-GTTGTCTCCTGCGACTTCA-3′;
*GAPDH* -reverse: 5′-TGGTCCAGGGTTTCTTACTC-3′.

**In situ hybridization.** Dissected mouse mammary glands were pre-fixed for 2 h in 4% PFA (Sigma) at room temperature, placed in PBS with 30% sucrose to minimize freeze fracture at 4 °C overnight, and then sectioned at 12 μm in a cryostat and collected on Superfrost PLUS slides. Slides were fixed in 4% PFA for 10 min at room temperature, washed twice in PBS, and immersed in 0.2 M HCl for 15 min, followed by two 5 min washes in PBS. Proteinase K (10 μg/ml) was added to the slides for 5 min at 37 °C. Slides were rinsed twice in 0.2% glycine, transferred to acetylation solution (590 ml DEPC-treated water, 8 ml triethanolamine, 1050 μl 37% HCl, 1.5 ml acetic anhydride), washed twice in PBS. For *miR-31 in situ* hybridization, digoxigenin-labeled LNA probes (Exiqon, Vedbaek, Denmark) were used following the manufacturer's protocol. TSA plus fluorescein system (PerkinElmer) was used to enhance signals.

**Dual luciferase activity assays.** The *Dkk1* 3′-UTR was amplified with the following primers: forward 5′-TA TGGGGAAGAGAAGAAACG-3′; and reverse 5′-TTTGGAAGGTATTGTC GGAA-3′. *Axin1* 3′-UTR was amplified with the following primers: forward 5′-GTTTCTTTTCAGCGGCACTC-3′; and reverse 5′-ATATTTACACGGACAC TTGG-3′). *Gsk3β* 3′-UTR was amplified with the following primers: forward 5′-TTAGCGGCCGCTCAGTTTCACAGGGTTAT-3′; and reverse 5′-GCGCTCGAGACAAAGGCATTCAAGTAG-3′). *Smad3* 3′-UTR was amplified with the following primers: forward 5′-CCGCTCGAGCACCA-CACCGAATGAATG-3′; and reverse 5′-ATAAGAATGCGGCCGCTGGCAATCC TTTACCATAGC-3′). *Smad4* 3′-UTR was amplified with the following primers:

forward 5′-TTACTCCTAGCAGCACCC-3′; and reverse 5′-CAGTTGTCG TCTTCCCTC-3′). 3′-UTRs were cloned into the XhoI and NotI sites of a modified psiCHECK2 vector (Promega) immediately downstream of the Renilla luciferase stop codon. Putative *miR-31* recognition elements in the *Dkk1* 3′-UTR were mutated by site-directed mutagenesis (Stratagene). For the dual luciferase assay, HC11 cells were transfected with 4 μg of *Dkk1* 3′-UTR, mutated 3′-UTR or control vector using Lipofectamine 2000 (Invitrogen). After 36 h, cells were lysed and the Dual-Glo luciferase reporter assay (Promega) was performed according to the manufacturer's protocol.

**TOP/FOP flash reporter assay**. HC11 mouse mammary epithelial cells stably transfected for *miR-31* inhibitor and Dox-inducible *miR-31* were transfected with 10 μg/100 μl TOPflash reporter plasmid or 10 μg/100μl FOPflash reporter plasmid using Lipofectamine 2000 (Invitrogen). The transfected cells were cultured in the absence or presence of Dox for 24 h, and both firefly and the renila luciferase activities were measured using Dual Luciferase Reporter Assay System Kit (Promega). TOP/FOP activities were calculated following the formula: TOP/FOP = (TOP firefly luciferase activity/renila luciferase activity)/(FOP firefly luciferase activity/renila luciferase activity). *MiR-31* inhibitor (anti-*miR-31*, GenePharma Co., Ltd), an artificial RNA sequence (5′-CAGCUAUGCCAGCA UCUUGCCU-3′). Scrambled RNA was used as a negative control (5′-CAG UACUUUUGUGUAGUACAA-3′). This experiment was repeated three times.

**Cell viability assays**. Cell viability was assessed by a tetrazolium salt (WST-8)-based colorimetric assay using the Cell Counting Kit 8 (CCK-8, Dojindo, Japan). Briefly, control and Dox-treated cells were seeded onto 96-well plates at an initial density of $5 \times 10^3$ cells per well. CCK-8 or medium was added to parallel sets of wells to control for CCK-8 interference. At specified time points, 10 μl of CCK-8 solution was added to each well of the plate, and the plate was incubated for 1 h at 37 °C. Cell viability was determined by scanning with a micro-plate reader at 450 nm. Data were expressed as the relative absorbance rate calculated as follows: relative viability = [A450(Dox)-A450(blank)]/ [A450(control)-A450(blank)]. For cell cycle analysis, cells were washed once in PBS, trypsinized, pelleted at 1,000×*g*, and rinsed once in 2 ml of cold PBS. After centrifugation, cells were slowly resuspended in 2 ml of cold 75% ethanol, after fixation for 2 h at 4 °C. Cells were washed in 1 ml PBS, stained with 50 μg/ml propidium iodide (PI, Sigma) and treated with 100 μg/ml RNase A (Invitrogen), 0.2% Triton X-100 (Sigma) for 30 min at 4 °C. Cells were analyzed for cell cycle stage by flow cytometry.

**Chromatin immunoprecipitation (ChIP) assay**. ChIP assays were performed using the SimpleCHIP enzymatic ChIP kit (Cell Signaling Technology) according to the protocol. Cells were harvested and cross-linked with 1% (v/v) formaldehyde for 10 min at RT. Then nuclei were isolated and chromatin was digested into fragments of 150–900 bp by micrococcalnuclease for 20 min at 37 °C, followed by ultrasonic disruption of the nuclear membrane using Sonics Vibra-Cell™ (80% amplitude, seven sets of 20 s pulses). The sonicated nuclear fractions were divided for input control and for overnight incubated at 4 °C with 5 mg of either anti-p65 Ab or the negative control IgG. After incubation with 30 μl of ChIP grade protein G-agarose beads for 2 h at 4 °C, the Ab-protein-DNA complexes were then eluted from the beads and digested by Proteinase K (40 mg) for 2 h at 65 °C, followed by purification of the DNA. Finally, genomic DNA recovered from the ChIP assays were qPCR amplified with primers. The primers used for detection of the p65-binding elements of the *miR-31* promoter sequence were as follows respectively: p65-binding site 1-forward: 5′-GCCATGCTTCTGTG-TAACCT-3′; p65 binding site 1-reverse: 5′-TCCTCACCTGTTATGCTTGTG-3′; p65 binding site 2-forward: 5′-GCAGTTCAAAGGGCAGTTCA-3′. p65 binding site 2-reverse: 5′-TGTGAGAACATCCCTGCACA-3′. The specificity of each primer set was verified by analyzing the dissociation curve of each gene-specific PCR product.

**BRCA subtype analysis**. For BRCA subtype analysis, mRNA expression levels from RNA-Seq on TCGA BRCA tumor samples were analyzed. Expression values were obtained by trimmed mean of M-values (TMM) normalization of read counts in genes by edgeR[67]. BRCA subtype was obtained by combining the classifications of PAM50[68] and hierarchical clustering of BRCA Expression values. We used the PAM50 classifier and its associated TCGA[69] training cohort by pamr[70] and then predicted subtype. Hierarchical clustering was analyzed with Pearson's correlation and average linkage.

**CLIP-qPCR assay**. CLIP-qPCR assay performed as previously described with modification. Cells were treated with scramble RNA or *miR-31* mimics, and then harvested after irradiated at 400 mJ/cm$^2$ for twice, then resuspended with PXL buffer including RNAsin (Promega) and RQ1 DNAse(Promega), 15,000 r.p.m. for 30 min, remaining supernatant. Protein A Dynabeads (Dynal, 100.02, Thermo Fisher, Fremont, CA) and goat anti-rabbit IgG (Jackson ImmunoResearch, West Grove, PA) or Ago2 antibody incubate for 4 h at 4 °C with rotation. The supernatant was added to the Beads for 2–4 h at 4 °C. Beads were washed twice and

digested with Proteinase K (4 mg/ml) after 20 min at 37 4 °C, RNA was extracted from beads using Trizol Reagent (invitrogen) and then quantified with qRT-PCR.

**Xenograft assay**. $5 \times 10^5$ ($n = 7$ mice each), $1 \times 10^3$ ($n = 6$ mice each), $1 \times 10^2$ ($n = 6$ mice each) PyVT and PyVT/KO primary breast cancer cells were injected into the fourth mammary fat pad of female NOD-SCID mice (6–8 weeks of age). Animals were sacrificed and analyzed 8 weeks after injection. 4T1 mouse mammary breast cancer cells were treated with *miR-31* inhibitor and scramble RNA (NC), then $5 \times 10^4$ inhibitor- and NC-treated cells were injected into the fourth mammary fat pad of female BALB/C mice (6–8 weeks of age). Animals were sacrificed and analyzed 4 weeks after injection. The mice were randomly divided into different groups before xenografting.

**Statistic analysis**. All analyses were performed in triplicate or greater. Two-tailed unpaired student's *t*-test was used for statistical analysis when a pair of conditions was compared. Asterisks denote statistical significance (*$P < 0.05$; **$P < 0.01$; ***$P < 0.001$). The data are reported as mean ± S.D., unless mean ± S.E.M. are specifically stated in figure legends. Survival curves were estimated by the Kaplan–Meier method and compared using the Wilcoxon test.

**Data availability**. The TCGA data referenced during the study are available in a public repository from the TCGA research website (https://cancergenome.nih.gov/). All other data are available within the Article and its Supplementary Files, or available from the authors upon request.

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

## Acknowledgements

We thank Bogi Andersen (University of California, Irvine) for editing the manuscript. Z.Y. is supported by the National Natural Science Foundation of China (No. 81572614, 31271584); Beijing Nature Foundation Grant (5162018); the Major Project for Cultivation Technology (2016ZX08008001, 2014ZX08008001); National Dairy Industry and Technology System (No.CARS-36); Basic Research Program (2015QC0104, 2015TC041, 2016SY001, 2016QC086); SKLB Open Grant (2015SKLB6-16). M.V.P. is supported by

the NIH NIAMS grants R01-AR067273, R01-AR069653. J.S. is supported by the National Natural Science Foundation of China (No. 31370830 and 11675134) and the 111 Project (No. B16029).

## Author contributions

C.L., F.L. and Z.Y. designed the experiment. C.L., F.L., X.L., Y.T., Y.Z., X.S., Q.M., S.Y., M.X. performed the experiments. C.L., F.L., X.L., Y.T., Y.Z., X.S., Q.M., S.Y., M.X. X.F., X.D., M.P., C.L., S.M., J.S., W.C. and B.J. analyzed the data. L.L. and T.A. generated the TRE-miR31 transgenic mice. Z.Y. and S.E.M. provided reagents. Z.Y., C.L., X.D., M.V.P., C.L., S.E.M., J.S., F.R. and W.C. wrote and edited the manuscript.

## Additional information

**Competing interests:** The authors declare no competing financial interests.

