## [Peer Review File · Nature Communications]

Reviewers' comments:

Reviewer #1 (Remarks to the Author):

The authors have addressed my concerns and I am supportive of publishing the revised manuscript.

Reviewer #2 (Remarks to the Author):

The authors have satisfactorily addressed all of my concerns

Reviewer #3 (Remarks to the Author):

The authors have largely addressed my previous comments. I think this is a viable candidate for Nature Communications and I accept the authors' response in certain cases where the requested experiment is not provided. However, I am concerned about the authors' response to the following points that I previously raised:

1. Regarding my previous point: "Individual mice can have big variations. For all mouse analyses, please specify the number of mice analyzed for each group, and if applicable, include data quantification and statistical analysis. For all Western blot analyses of mouse tissues (e.g., Fig. 2o, 3h, 5c, 5g, 6c, and 7h), multiple mice per group should be used", the authors ignored the need to use multiple mice per group for Western blot analyses. One mouse per group cannot get conclusive results.

2. Regarding my previous point: "The genetic background (B6? FVB?) of all mutants should be specified, because the mouse strain profoundly affects breast cancer phenotypes." The authors stated that the miR-31 mutants are B6 and the MMTV-PyVT mice are FVB, i.e., they used mixed background for the compound mutants, instead of pure background. However, it is well known that for the MMTV-PyVT mice, the tumor and metastasis phenotypes of the FVB strain is much more aggressive than the B6 strain. Therefore, it is critical to compare mice with the identical genetic background in this study. Mice of mixed background do not always have the same genetic makeup (unless they are littermates).

Response to Reviewer Comments:

Reviewer #1 (Remarks to the Author):

The authors have addressed my concerns and I am supportive of publishing the revised manuscript.

We thank Reviewer #1 for his/her enthusiasm and insightful comments on our work!

Reviewer #2 (Remarks to the Author):

The authors have satisfactorily addressed all of my concerns

We thank Reviewer #2 for his/her enthusiasm and insightful comments on our work!

Reviewer #3 (Remarks to the Author):

The authors have largely addressed my previous comments. I think this is a viable candidate for Nature Communications and I accept the authors' response in certain cases where the requested experiment is not provided. However, I am concerned about the authors' response to the following points that I previously raised:

We thank Reviewer #3 for his/her enthusiasm and insightful comments on our work! We have addressed all of the remaining concerns below.

1. Regarding my previous point: "Individual mice can have big variations. For all mouse analyses, please specify the number of mice analyzed for each group, and if applicable, include data quantification and statistical analysis.

We thank the reviewer for this important suggestion. In the revised manuscript, we have explicitly specified the numbers of mice for each experiment in the Figure legends. The statistical analysis has been included in the Figure legends and in the Method section.

For all Western blot analyses of mouse tissues (e.g., Fig. 2o, 3h, 5c, 5g, 6c, and 7h), multiple mice per group should be used", the authors ignored the need to use multiple mice per group for Western blot analyses. One mouse per group cannot get conclusive results.

We would like to clarify that for the original submission we performed three biological replicates for each Western blot analysis, however, we only showed one representative image from each pair of mice. To comprehensively address this question, we re-collected protein samples and performed new Western blot analysis with three controls and three experimental samples. We are now showing new Western blot analysis results in the revised **Fig. 2o, 39, 3i, 3o, 5f, 6h, 7c, 7g, 8c and 8g**.

2. Regarding my previous point: "The genetic background (B6? FVB?) of all mutants should be specified, because the mouse strain profoundly affects breast cancer phenotypes." The authors stated that the miR-31 mutants are B6 and the MMTV-PyVT mice are FVB, i.e., they used mixed background for the compound mutants, instead of pure background. However, it is well known that for the MMTV-PyVT mice, the tumor and metastasis phenotypes of the FVB strain is much more aggressive than the B6 strain. Therefore, it is critical to compare mice with the identical genetic background in this study. Mice of mixed background do not always have the same genetic makeup (unless they are littermates).

Indeed, this is a good point. For this study, we used littermates to make any comparison between *MMTV-PyVT* and *MMTV-PyVT/KO* mice. For the mammary gland phenotype analysis, we also used littermates as controls for the KO, cKO and DTG mice.

REVIEWERS' COMMENTS:

Reviewer #3 (Remarks to the Author):

The authors have addressed all my concerns.

Response to Reviewer Comments:

REVIEWERS' COMMENTS:

Reviewer #3 (Remarks to the Author):
The authors have addressed all my concerns.

We thank Reviewer #3 for his/her enthusiasm and insightful comments on our work!